# Self-cycled photo-Fenton-like system based on an artificial leaf with a solar-to-$H_2O_2$ conversion efficiency of 1.46%

Chaoran Dong[1], Yilong Yang[1], Xuemin Hu[1], Yoonjun Cho[2], Gyuyong Jang[2], Yanhui Ao[3] ✉, Luyang Wang[4], Jinyou Shen[5], Jong Hyeok Park [2] ✉ & Kan Zhang [1] ✉

Millions of families around the world remain vulnerable to water scarcity and have no access to drinking water. Advanced oxidation processes (AOPs) are an effective way towards water purification with qualified reactive oxygen species (ROSs) while are impeded by the high-cost and tedious process in either input of consumable reagent, production of ROSs, and the pre-treatment of supporting electrolyte. Herein, we couple solar light-assisted $H_2O_2$ production from water and photo-Fenton-like reactions into a self-cyclable system by using an artificial leaf, achieving an unassisted $H_2O_2$ production rate of 0.77 μmol/(min·cm²) under 1 Sun AM 1.5 illumination. Furthermore, a large (70 cm²) artificial leaf was used for an unassisted solar-driven bicarbonate-activated hydrogen peroxide (BAP) system with recycled catalysts for real-time wastewater purification with requirements for only water, oxygen and sunlight. This demonstration highlights the feasibility and scalability of photoelectrochemical technology for decentralized environmental governance applications from laboratory benchtops to industry.

Advanced oxidation processes (AOPs) are a recognized approach that can effectively treat various organic pollutions in aqueous solutions because the generated reactive oxygen species (ROSs) have strong oxidation capabilities for mineralizing the organic molecules[1]. Fenton reaction is one of the most representative AOPs which can rapidly generate hydroxyl radicals (·OH) with high concentration, therefore being widely adopted in wastewater treatment engineering[2]. However, the Fenton reaction is an irreversible process that needs the continuous supply of various chemical reagents, such as $H_2O_2$, iron salt, acid/alkali et al., which increases the costs of the whole wastewater treatment process by more than one time[3]. Heterogeneous photocatalysis over semiconductors can generate a range of ROSs by

photoinduced redox processes, which are regarded as green and sustainable AOPs with affordable cost[4]. In the past several decades, numerous efforts have been devoted to photocatalytic degradation of organic pollution by developing or designing various photocatalysts and photocatalytic systems[1,5]. However, directly generating ROSs by photoinduced redox processes in aqueous solutions are profoundly restricted by rapid recombination of photoinduced electron/hole pairs, and high thermodynamic barriers of radical generation (e.g., $\cdot O_2^-$ at −0.33 V and ·OH at 1.99 V vs. normal hydrogen electrode, NHE), while combined photocatalysis with Fenton reaction, so-called photo-Fenton reaction, also requires the supply of $H_2O_2$[6,7]. Therefore, there is a great necessity to find indeed efficient, low-cost, and sustainable AOPs.

[1]MIIT Key Laboratory of Advanced Display Material and Devices, School of Materials Science and Engineering, Nanjing University of Science and Technology, 210094 Nanjing, China. [2]Department of Chemical and Biomolecular Engineering, Yonsei University, 50 Yonsei-ro, Seodaemun-gu, Seoul 120-749, Republic of Korea. [3]Key Laboratory of Integrated Regulation and Resource Development on Shallow Lakes of Ministry of Education, College of Environment, Hohai University, 210098 Nanjing, China. [4]College of New Materials and New Energies, Shenzhen Technology University, Shenzhen, Guangdong, P. R. China. [5]Jiangsu Key Laboratory of Chemical Pollution Control and Resources Reuse, School of Environmental and Biological Engineering, Nanjing University of Science and Technology, 210094 Nanjing, China. ✉e-mail: andyao@hhu.edu.cn; lutts@yonsei.ac.kr; zhangkan@njust.edu.cn

In recent years, it is found that bicarbonate ($HCO_3^-$) can not only react with $H_2O_2$ to form $HCO_4^-$ with high reactivity but also act as an auxiliary for water-oxidative $H_2O_2$ evolution under photoelectrochemical (PEC) condition, through a reversible reaction as follows[8,9]:

$$HCO_3^- + H_2O_2 \leftrightarrow HCO_4^- + H_2O \tag{1}$$

The above reversible reaction offers a possibility of an in situ generation and decomposition of $H_2O_2$ in an aqueous solution containing $HCO_3^-$ under solar light illumination[10–12]. In addition, the co-existence of $HCO_4^-$ and $H_2O_2$ was reported to facilitate the formation of various ROSs ($\cdot O_2^-$, $\cdot OH$, $CO_3^{\cdot-}$ et al.) by Co, Cu or Mn ion activation[13–15]. The correlation between $HCO_3^-$, $H_2O_2$, $HCO_4^-$ and free radical species inspire us to design a self-cycled AOPs system with high efficiency and affordable cost[16–18].

Herein, we propose a self-cycled Fenton-like system based on the deliberately designed artificial leaf, realizing the sustainable reaction system for wastewater treatment (Fig. 1a). First, the solar-motivated unassisted artificial leaf comprised of a $SnO_{2-x}$/$BiVO_4$/$WO_3$ photoelectrode and a poly tetrafluoroethylene (PTFE)-modified Mo single-atom catalysts/mildly reduced graphene oxide-coated gas diffusion electrode (PTFE@Mo-SACs/mrG-GDE) cathode fulfills the efficient $H_2O_2$ production in a bicarbonate-containing electrolyte with a production rate of 0.77 µmol/(min cm$^2$) under AM 1.5 G illumination, corresponding to a bias-free solar-to-hydrogen-peroxide efficiency (SHyE) of 1.46%. Second, the generated $H_2O_2$ can be immediately in situ activated into main $\cdot OH$, $\cdot O_2^-$ and $^1O_2$ et al. via the catalysis of Mn(II) species in the bicarbonate electrolyte, and the Mn(II) species are correspondingly oxidized to high-valent Mn(IV) species. Third, the Mn species can be recycled by reducing the Mn(IV) species into Mn(II) in the cathode (Fig. 1a). As a result, the self-cycling process with requirements for only water, oxygen, and sunlight demonstrates long-term stability for more than one month for the removal of various organic pollutants.

## Results

To demonstrate the self-cycled AOPs, the formation of ROSs and redox coupling of Mn(II)/Mn(IV) are first evidenced. As shown in Supplementary Fig. 1, the electron paramagnetic resonance (EPR) spectra of the co-existence of $HCO_3^-$, $H_2O_2$, and Mn(II) demonstrate the remarkably enhanced signal intensities of $\cdot OH$, $\cdot O_2^-$ and $^1O_2$ compared to those of the $HCO_3^-$/$H_2O_2$ or $H_2O_2$/Mn(II), indicating that multi-species of ROSs can be activated by the self-cycled AOPs system[19]. Then, both cyclic voltammetry (CV) tests and EPR spectroscopy were used to track the Mn species. As shown in

Supplementary Fig. 2, electrochemical redox peaks relative to Mn(II)/Mn(III) and Mn(III)/Mn(IV) appear in the potential window between 0.98 and 1.58 V vs. reversible hydrogen electrode (RHE), the narrower potential window would be easily met by photoinduced charge. Further insight into the valence alternation of Mn species during the redox is revealed by EPR measurement. As shown in Fig. 1b, a typical Mn(II) signal with a signature six-line pattern centered around g-2 is observed in the initial 2 min (curve i)[15]. However, both the symmetry and amplitude of this signal gradually decreased in 10 min (curve ii), ascribing to the Mn(II) being oxidized when the $H_2O_2$ is in situ generated. Accordingly, a new signal around g-4 corresponding to Mn(IV) species is observed, confirming the Mn(II) being oxidized to Mn(IV)[20]. Furthermore, the Mn(II) signals are barely observed after the light off for 10 min (curve iii) due to its consumption during the activation of $H_2O_2$ as well as the blocking of the regeneration of Mn(II) caused by the suspending of counterpart reduction reaction in cathode[21,22]. Significantly, the oxidation/reduction quasi-equilibrium reappears after reopening the light (curve iv), indicting the regeneration of Mn(II) from the Mn(IV) reduction under the photo-activated cathode[20]. As a result, it is unambiguously confirmed that the Mn species can be recycled as a "redox shuttle" under the solar-activated self-cycled system. In order to further evaluate the vital role of the self-cycled system, control experiments were conducted in the H-cell that separates the cathode and Mn species. As shown in Supplementary Fig. 3a, under continuous light illumination, the signal of Mn(IV) only appears in the H-cell, while a stable quasi-equilibrium of Mn(II)/Mn(IV) in the one-compartment opened reactor (Supplementary Fig. 3b).

To realize the on-site $H_2O_2$ production, the PEC water-oxidative $H_2O_2$ evolution and electrochemical oxygen-reductive $H_2O_2$ reaction are simultaneously employed. According to our previous reports, a $SnO_{2-x}$-coated $BiVO_4$ ($SnO_{2-x}$/$BiVO_4$) photoanode can kinetically favor the $2e^-$ water oxidation reaction ($2e^-$WOR) for $H_2O_2$ evolution accompanied by slight water-oxidative $\cdot OH$ generation by $1e^-$WOR in a $NaHCO_3$ electrolyte because the $SnO_{2-x}$ overlayer tunes the solid/liquid energetics[23], and Mo-SACs/mrG, having the low thermodynamic barrier of $2e^-$ oxygen reduction reaction ($2e^-$ORR), can present high activity and selectivity for electrochemical $H_2O_2$ generation in a $NaHCO_3$ electrolyte[24]. Here, to drive the $2e^-$ORR by solar light only, a $SnO_{2-x}$/$BiVO_4$/$WO_3$ photoanode is designed since the $BiVO_4$/$WO_3$ heterojunction is well-demonstrated to lower onset potential and increase photocurrent density[25]. Figure 2 shows a schematic diagram of the solar-driven $H_2O_2$ production device using a $SnO_{2-x}$/$BiVO_4$/$WO_3$ photoanode and a Mo-SACs/mrG cathode. By connecting the water-oxidative generation of $H_2O_2$ at the photoanode with the $O_2$-reductive

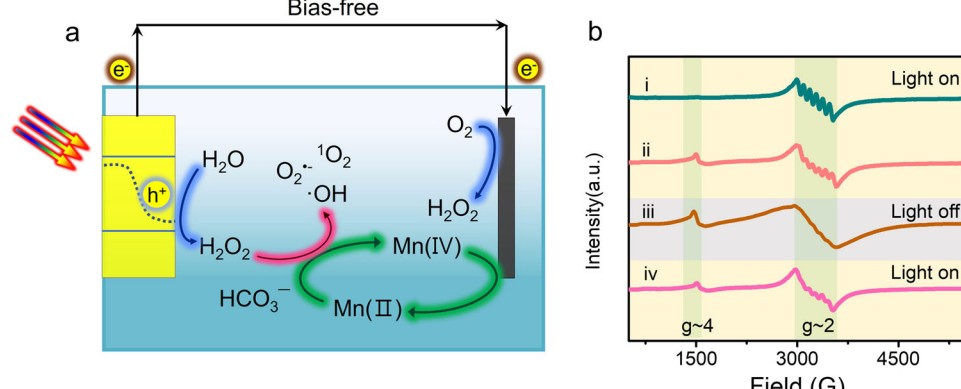

**Fig. 1 | Self-cycled photo-Fenton-like system with recycled Mn species.**
**a** Scheme of the Self-cycled photo-Fenton-like system with recycled Mn species.
**b** EPR response of Mn species: (i) Mn(II) signal in the initial 2 min under AM 1.5 illumination. (ii) Mn(II)/Mn(IV) signal in the initial 10 min under AM 1.5 illumination. (iii) Mn(II)/Mn(IV) signals after 5 min since the light off. (iv) Mn(II)/Mn(IV) signals after 5 min since the light reopened.

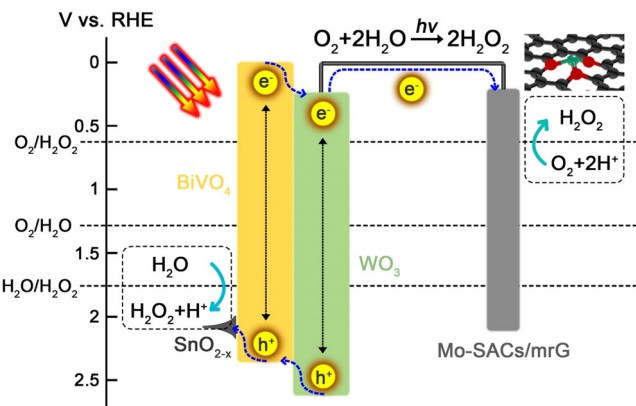

**Fig. 2 | Energy-level diagram of the photoanode and cathode: type 2 heterostructures formed between BiVO₄/WO₃.** Photo-induced electron-hole pairs were separated via the inner electronic field; the electron transferred to the single-metal sites in the cathode through the external circuit to complete the 2e⁻ORR reaction, while the hole transferred to the BiVO₄ surface and participated in the 2e⁻WOR on the SnO₂₋ₓ/BiVO₄/WO₃ photoanode.

generation of $H_2O_2$ at the cathode, the photovoltage and photocurrent of the coupled photochemical reaction are constrained to be equal at the thermodynamic level. The redox potential of the BiVO₄/WO₃ photoanode electrode is 0.34 V vs. RHE and 2.45 V vs. RHE[26], which straddle the oxidation potential of the 2e⁻ WOR (1.77 V vs. RHE) and the reduction potential of the 2e⁻ORR (0.70 V vs. RHE), which means that charge carriers excited by solar light can have enough energy to activate the overall reaction[27]. The operating point for such a coupled photochemical reaction can be established at the intersection point of the independently measured J−V curves. As shown in Supplementary Fig. 4, the point of intersection corresponds to the operating current density, $J_{OP}$. Considering the possible nonlinear correlation between the pH and redox potential, the SHyE should be calculated using $H_2O_2$ production rate ($r(H_2O_2)$) and the Gibbs free energy from $O_2$ and $H_2O$ to $H_2O_2$ ($\Delta G(H_2O_2)$) and sunlight power intensity ($P_{sun}$)[28]:

$$SHyE = \frac{r(H_2O_2) \times \Delta G(H_2O_2)}{P_{sun}} \quad (2)$$

Either increasing the reaction kinetics (slope of the J−V curve) or cathodically shifting the onset potential of the photoanode or anodically shifting the onset potential of the cathode raises the intersection point of the J−V curve of the photoanode and cathode, corresponding to a considerable increase in the working current of the coupled photochemical reaction under bias-free conditions[29], which is the key to assisting AOPs with desirable $H_2O_2$ concentration.

On the basis of the above discussions, both photoanode and cathode are rationally optimized to achieve high SHyE. By tailoring the WO₃ texture (Supplementary Figs. 5 and 6) and BiVO₄ content (Supplementary Figs. 7 and 8), the photocurrent density is first optimized to 5.63 mA/cm² at 1.76 V vs. RHE with an onset potential of 0.34 V vs. RHE. Then, a similar SnO₂₋ₓ overlayer approach was adopted to increase the Faradic efficiency (FE) of $H_2O_2$ production[23], as schematically illustrated in Fig. 3a. The optimized SnO₂₋ₓ/BiVO₄/WO₃ was characterized by a combination of techniques, which indicate a porous mesoscopic photoelectrode with a microscopic layer-by-layer component structure consisting of ~5 nm thick SnO₂₋ₓ on a porous BiVO₄/WO₃ film (Fig. 3b−d and Supplementary Figs. 9 and 10). UV−vis diffuse reflectance spectra reveal that SnO₂₋ₓ/BiVO₄/WO₃ has an absorption edge similar to that of BiVO₄/WO₃ but much greater than the absorption edge of WO₃ (Supplementary Fig. 11), suggesting a negligible influence on the energy-level diagram of the SnO₂₋ₓ overlayer on

BiVO₄/WO₃. Electrochemical impedance spectroscopy (EIS) spectra demonstrate a large improvement in the charge carrier transport efficiency of SnO₂₋ₓ/BiVO₄/WO₃ compared to BiVO₄ (Fig. 3e). The PEC water-oxidative $H_2O_2$ generation performance measured in 0.4 M NaHCO₃ under 100 mW/cm² with AM 1.5 illumination shows that the SnO₂₋ₓ overlayer slightly enhances the photocurrent density of BiVO₄/WO₃ (6.1 mA/cm² at 1.76 V vs. RHE), while cathodically shifting the onset potential from 0.64 V vs. RHE of BiVO₄ to 0.32 V vs. RHE (Fig. 3f). Significantly, the SnO₂₋ₓ/BiVO₄/WO₃ photoanode exhibits a stable $H_2O_2$ FE greater than 84% at potentials ranging from 0.5 to 1.7 V vs. RHE, which is 2.4 and 2.9 times higher than those of BiVO₄/WO₃ and BiVO₄, respectively (Fig. 3g). The potential-independent $H_2O_2$ FE is a notable characteristic feature of the PEC reaction because the rate constant and activation energy are less affected at the semiconductor/electrolyte interface[30,31], which is important for achieving a high SHyE in the coupled photochemical reaction. Furthermore, potentiostat $i−t$ tests reveal that the SnO₂₋ₓ/BiVO₄/WO₃ photoanode is more stable than BiVO₄/WO₃ and BiVO₄ (Supplementary Fig. 12) because the passivation layer inhibits the photocorrosion of BiVO₄[32]. As mentioned above, the higher selectivity of water-oxidative $H_2O_2$ generation together with high photocurrent density, low onset potential, and good stability are key factors in achieving high SHyE since the coupled photochemical reaction is mainly driven by the photoanode here.

It is essential to realize that the water-oxidative generation of $H_2O_2$ at the SnO₂₋ₓ/BiVO₄/WO₃ photoanode accompanies the water-reductive generation of $H_2$ at the Pt cathode, where the overpotential of the HER is almost perfectly neat. Compared to that of water splitting to form $H_2O_2$ and $H_2$, the actual voltage required to drive the coupled photochemical reaction of $H_2O_2$ generation is greater and can be expressed by:

$$V = 1.07 V + \eta_{WOR} + \eta_{ORR} + \eta_R \quad (3)$$

where $\eta_{WOR}$ is the overpotential of the WOR, $\eta_{ORR}$ is the overpotential of the ORR and $\eta_R$ is the inner resistance of the PEC cell. Therefore, decreasing $\eta_{ORR}$ (given by the Tafel behavior of ORR catalyst) is essential to achieve high $H_2O_2$ production rates and SHyE. First, Mo-SACs/mrG was carefully chosen as the cathode catalyst for oxygen-reductive $H_2O_2$ production[24]. Rotating ring-disk electrode (RRDE) tests were carried out for in situ quantification of the $H_2O_2$ catalytic activity of the Mo-SACs/mrG catalysts. To exclude the impact of the carbonaceous substrate, which is proposed to serve as a potential catalyst toward electrochemical 2e⁻ORR[33], a control sample denoted mrG was prepared via the same fabrication process used for Mo-SACs/mrG but without a metal source. Distinct RRDE curves of Mo-SACs/mrG and mrG are presented in Fig. 4a. The Tafel slope and onset potential (at a disk current density of 1 mA/cm²) of Mo-SACs/mrG are 81 mV/dec and 0.61 V vs. RHE, respectively, while those of mrG are 190 mV/dec and 0.55 V vs. RHE, respectively (Fig. 4a and Supplementary Fig. 13). Furthermore, a much higher $H_2O_2$ selectivity was observed for Mo-SACs/mrG compared to mrG from 0.35 to 0.65 V vs. RHE (Fig. 4b). Therefore, Mo-SACs/mrG was concluded to enable optimization of the intersection point of the J−V curve of the coupled photoanode and cathode with ideal $H_2O_2$ FE.

Prior to physically coupling the cathode and photoanode, the charge balance between the cathode and photoanode should be optimized since the activity of the cathode is limited by gas diffusion and electron transfer[34]. Occasionally, a microporous gas diffusion layer (GDL) composed of PTFE and carbon-black powder is introduced between the active catalysts and carbon fiber to enhance the gas diffusion efficiency and boost the three-phase contact line (TPCL), on which the accessible active sites are functional. Nevertheless, a thick and insulated GDL will slow down electron transfer, resulting in serious ohmic loss and performance decay[35]. Therefore, increasing the TPCL without hindering electron transfer is desirable. Herein, an interfacial

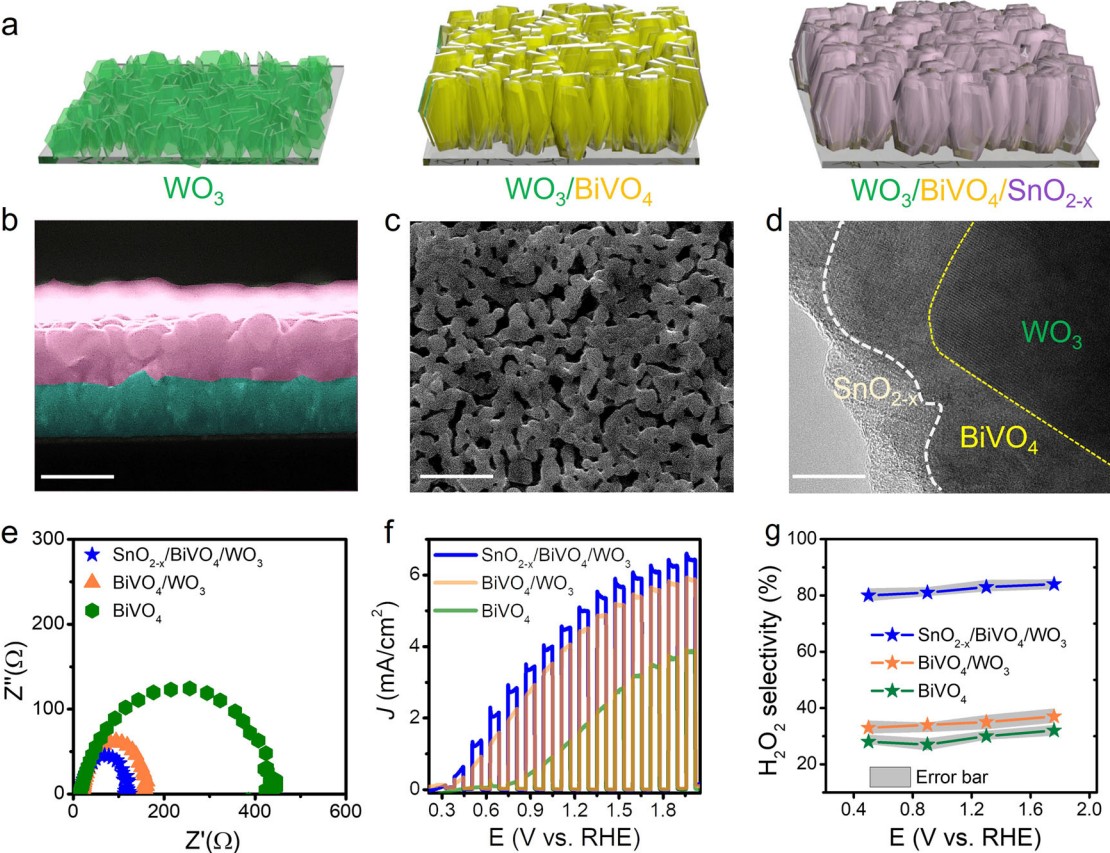

**Fig. 3 | Multiheterostructure photoanode for effective $H_2O_2$ production.**
**a** Scheme of the construction of the heterostructure of $SnO_{2-x}/BiVO_4/WO_3$. **b** SEM cross-sectional image of $SnO_{2-x}/BiVO_4/WO_3$. Scale bar: 500 nm. **c** SEM surface image of $SnO_{2-x}/BiVO_4/WO_3$. Scale bar: 500 nm. **d** HR-TEM image of $SnO_{2-x}/BiVO_4/WO_3$, Scale bar: 5 nm. **e** EIS scan of $BiVO_4$, $BiVO_4/WO_3$, and $SnO_{2-x}/BiVO_4/WO_3$ under AM 1.5 illumination in 0.4 M $NaHCO_3$ electrolyte. **f** Photocurrent linear sweep voltammetry (LSV) scan of $BiVO_4$, $BiVO_4/WO_3$ and $SnO_{2-x}/BiVO_4/WO_3$ under AM 1.5 illumination. **g** $H_2O_2$ FE values of $BiVO_4$, $BiVO_4/WO_3$, and $SnO_{2-x}/BiVO_4/WO_3$ with controlled potential under AM 1.5 illumination in 0.4 M $NaHCO_3$ electrolyte.

strategy is proposed to construct an oxygen local confinement structure on Mo-SACs/mrG nanosheets via PTFE nanoparticle decoration to increase the TPCL without inducing GDL (Fig. 4c and d). The PTFE@Mo-SACs/mrG-GDE cathode was characterized by SEM and HAADF-STEM, and the results are shown in Supplementary Fig. 14. The mrG nanosheets adhere tightly to the carbon fiber array, and PTFE nanoparticles with a size of ~150 nm are evenly distributed on the surface of mrG nanosheets, where single Mo atoms are the only metal species present (Supplementary Fig. 15). As illustrated in Supplementary Fig. 16, the quasi-nanoarray aerophilic area generated by the PTFE nanoparticle combined with Mo-SACs/mrG nanosheets is able to simultaneously contribute to compatible TPCL and significantly enhanced electron transfer compared to the conventional GDL electrode[36,37]. Further insights into the influence of PTFE on the electrocatalytic activity are obtained by altering the loading amount of the PTFE nanoparticles (Supplementary Fig. 17). Water contact angle tests reveal that the wettability of PTFE@Mo-SACs/mrG-GDE could be efficiently suppressed when the loading amount of PTFE increased (Supplementary Fig. 18), which is expected to accelerate gas diffusion. However, excessive PTFE loading deteriorates the conductivity of the electrode, as evidenced by EIS (Supplementary Fig. 19). Therefore, a delicate balance among the mass transfer of oxygen, the conductivity of the electrode and $H_2O_2$ formation efficiency is required for performance optimization. Ultimately, a 2:1 mass ratio of PTFE and Mo-SACs/mrG achieves the best performance, with a current density of 10 mA/cm² at 0.51 V vs. RHE and a Tafel slope of 53.2 mV/dec (Supplementary Fig. 20). As a result, PTFE@Mo-SACs/mrG-GDE shows the highest current density in the voltage range from

0.65 to 0.1 V vs. RHE (Fig. 4c), with the smallest onset potential (0.51 V vs. RHE) and the largest $H_2O_2$ FE (82.1%) (Fig. 4e) compared to the Mo-SACs/mrG-GDL electrode (see "Methods" for synthesis) and PTFE@mrG-GDE (see "Methods" for synthesis), indicating that both the oxygen-confined structure and the effective Mo-SACs/mrG catalyst contribute to the outstanding electrochemical performance. Notably, the optimized PTFE@Mo-SACs/mrG-GDE cathode exhibits good electrochemical stability for the ORR, with 98.4% of the initial current maintained after 7 h (Supplementary Fig. 21).

After optimization of the $H_2O_2$ product selectivity, onset potential, and current density, the assembled $SnO_{2-x}/BiVO_4/WO_3$||PTFE@Mo-SACs/mrG-GDE PEC cell for redox coupling of the $2e^-$ WOR and $2e^-$ORR has a theoretical intersection point at 0.61 V vs. RHE at a current density of 1.64 mA/cm² (Fig. 5a). The PEC performance was first investigated in a three-electrode system with $SnO_{2-x}/BiVO_4/WO_3$ and PTFE@Mo-SACs/mrG-GDE as the photoanode and cathode, respectively. The linear sweep voltammetry (LSV) scan reveals a current density of 5.59 mA/cm² at 1.76 V vs. RHE with a total $H_2O_2$ FE of 151% calculated from both the photoanode and cathode (Supplementary Fig. 22). Next, controlled potential tests were carried out to evaluate the $H_2O_2$ FE of the PEC device in a two-electrode system to evaluate the monolithic cell performance, and the results demonstrate a steady FE greater than 150% under a wide applied voltage range from 0 to 0.9 V vs. cell (Fig. 5b). Specifically, a bias-free photocurrent density of 1.56 mA/cm² with an $H_2O_2$ FE of 152% was achieved (Fig. 5c), corresponding to an unassisted $H_2O_2$ production rate of 0.77 μmol/(min cm²) and an SHyE of 1.46%. The difference from the theoretical value is attributed to the PEC limitation, such as the pH gradient, IR drop, and

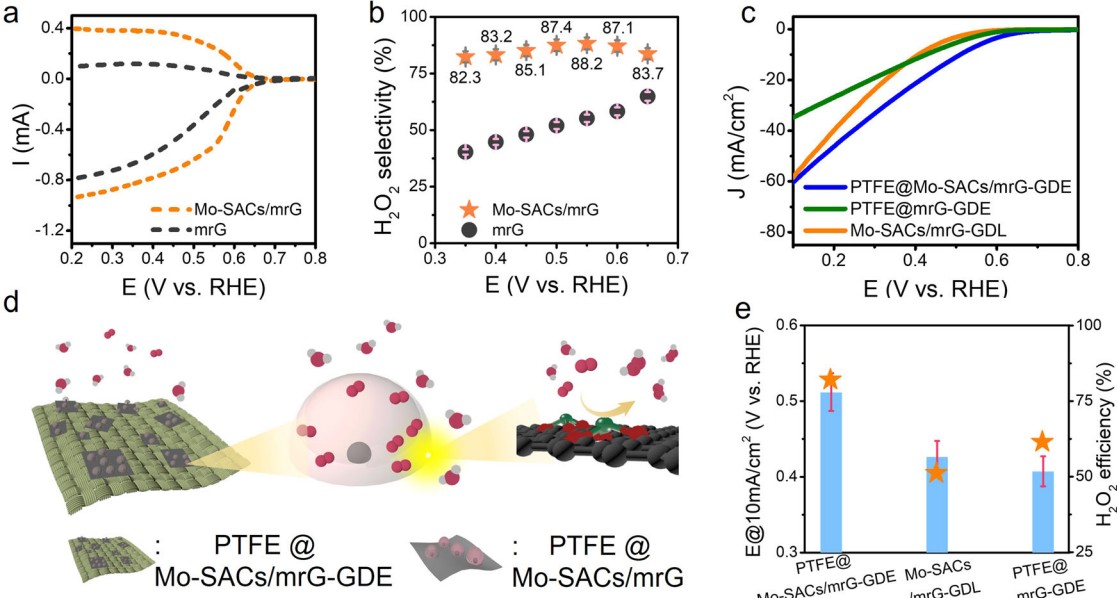

**Fig. 4 | Catalysts and oxygen local confinement structure of the cathode for H₂O₂ production. a** RRDE tests of Mo-SACs/mrG under O₂-purged 0.4 M NaHCO₃ electrolytes. **b** FE of H₂O₂ productions over Mo-SACs/mrG obtained by RRDE tests at various applied voltages. **c** LSV scan of the PTFE@Mo-SACs/mrG-GDE, PTFE@mrG-GDE and Mo-SACs/mrG-GDL electrodes under an O₂-purged 0.4 M NaHCO₃ electrolyte. **d** Scheme of the structure of the PTFE@Mo-SACs/mrG-GDE electrode. **e** Potential at 10 mA/cm² and corresponding H₂O₂ FE of PTFE@Mo-SACs/mrG-GDE, PTFE@mrG-GDE and Mo-SACs/mrG-GDL electrodes. The pentagram represents the H₂O₂ efficiency, and the blue rectangle represents the onset potential at 10 mA/cm².

the polarization that occurs at the counter electrode in the two-electrode system ($\eta_R$ in Eq. (2))[34,38]. As a result, an average solar-driven H₂O₂ production rate of 0.74 µmol/(min cm²) is achieved, and the H₂O₂ concentration reached 0.85 mM within one hour in a 50 mL reactor (Fig. 5d). A summary of the PEC and solar-driven H₂O₂ production exhibits the highest unassisted solar-driven H₂O₂ production rate of our SnO₂₋ₓ/BiVO₄/WO₃||PTFE@Mo-SACs/mrG-GDE PEC device (Fig. 5e and Supplementary Table 1), surpassing other state-of-the-art PEC systems[23,39–49]. Potentiostatic *i–t* tests were carried out under bias-free conditions and showed good stability of the PEC device, with 89% of the current retained after 20 h of H₂O₂ generation (Supplementary Fig. 23).

As a demonstration, a large SnO₂₋ₓ/BiVO₄/WO₃||PTFE@Mo-SACs/mrG-GDE monolithic artificial leaf (7 cm²) was engineered to enable a wastewater purification system with adequate H₂O₂ output (Fig. 6a). As shown in Fig. 6b, a significant improvement in current is observed when the areas of both the SnO₂₋ₓ/BiVO₄/WO₃ photoanode and PTFE@Mo-SACs/mrG-GDE cathode are increased to 7 cm², which yields a theoretical intersection point at 0.55 V vs. RHE and a current of 12.3 mA, indicating the good scalability of the SnO₂₋ₓ/BiVO₄/WO₃||PTFE@Mo-SACs/mrG-GDE PEC device. As a result, an unassisted current of 10.7 mA with a H₂O₂ FE of 150% and a production rate of 5.17 µmol/min is achieved (Fig. 6c). Next, as a demonstration of a PEC-driven bicarbonate activated hydrogen peroxide(BAP) system, the artificial leaf is applied for the degradation of 4-nitrophenol (NP), which is considered as a priority toxic pollutant by U.S. Environmental Protection Agency (EPA) and exhibits particular resistance to chemical/biological oxidation due to the electron-withdrawing nitro group[50]. As shown in Fig. 6d, 30% of NP (10 ppm) can be removed in 90 min without any Mn(II), whereas negligible degradation can be observed in absence of the artificial leaf. On the other hand, 99.5% NP is degraded within 50 min when only 0.04 ppm Mn(II) was introduced (lower than the maximum contamination for drinking water set by the U.S. EPA), suggesting the curial roles of the artificial leaf, H₂O₂ and Mn(II) in the BAP system. Further insight into the feasibility of the PEC-driven BAP system is obtained by assessing the influence of the

concentration of H₂O₂, Mn(II) species, and electrolyte. As shown in Supplementary Fig. 24a, a contrast artificial leaf sample was engineered by replacing the photoanode and cathode with SnO₂₋ₓ/BiVO₄ and Mo-SACs/mrG–GDE, respectively. The sharply decreased unassisted current density of 0.55 mA/cm² results in a critically reduced degradation rate that only 20% of 4-NP can be removed within 50 min in SnO₂₋ₓ/BiVO₄||Mo-SACs/mrG–GDE based artificial leaf, which is much lower than that of 99.5% by using the SnO₂₋ₓ/BiVO₄/WO₃||PTFE@Mo-SACs/mrG-GDE based artificial leaf (Supplementary Fig. 24b). This result suggests the determining role of higher H₂O₂ production rate towards the degradation efficiency. Furthermore, the degradation effect sharply deteriorates when the Mn(II) content reaches 1 ppm (Supplementary Fig. 25). It is supposed that excessive Mn(II) species might decompose the H₂O₂ into H₂O and O₂ rather than activate it into ROSs. Besides, it is evidenced that no Mn species are being deposited in the photoanode according to the XPS result (Supplementary Fig. 26). Either excessive or insufficient bicarbonate concentrations show a negative effect on the degradation efficiency (Supplementary Fig. 27). It can be tentatively ascribed to that the Mn ions tend to precipitate into inactive MnCO₃ when the bicarbonate concentration is too high[15]. In contrast, an excessively low bicarbonate concentration could reduce the photocurrent of the photoanode, which reduces the H₂O₂ production rate, thereby decreasing the degradation rate. Furthermore, a stability evaluation of the BAP system was carried out by repeated degradation tests, which shows hardly any deterioration in the degradation efficiency (Supplementary Fig. 28). To further demonstrate the practical applications of the PEC-driven BAP system, especially, the area of the artificial leaf is increased to 7 × 10 cm² for the effective degradation of a 200 mL synthetic sample containing 5 ppm rhodamine B (Rh. B), 5 ppm methylene blue (MB) and 5 ppm NP (Supplementary Fig. 29 a). As shown in Supplementary Fig. 29b, c, 99.5% of the pollutants can be removed within 70 min with only light and oxygen supplementation (Supplementary Movie 1). No obvious activity deterioration was observed after one month, demonstrating the robustness and long-term stability of the artificial leaf (Supplementary Fig. 30). Further measurements are

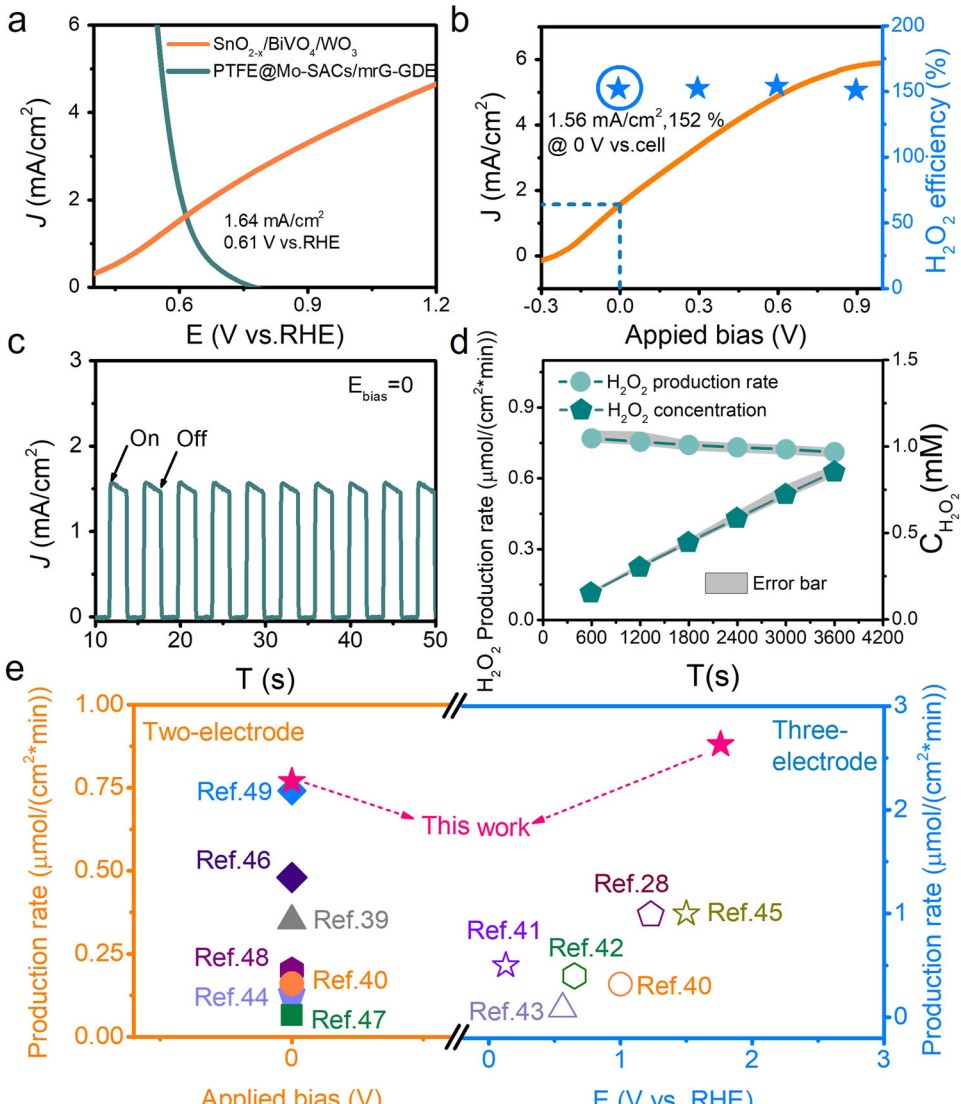

**Fig. 5 | Coupling photoanode and cathode for solar-driven $H_2O_2$ production and summary. a** Coupled LSV scan of the $SnO_{2-x}/BiVO_4/WO_3$ photoanode and PTFE@Mo-SACs/mrG-GDE cathode. **b** LSV scan of $SnO_{2-x}/BiVO_4/WO_3$||PTFE@Mo-SACs/mrG-GDE under a two-electrode system and their $H_2O_2$ FE at different applied voltages. **c** Photocurrent–time profiles under bias-free conditions. **d** $H_2O_2$ production rate and accumulated concentration of a one-cell configuration PEC device under bias-free conditions. **e** Summary of PEC $H_2O_2$ production under AM 1.5 illumination with both two-electrode and three-electrode systems. The electrolyte from **a**–**d** is 0.4 M $NaHCO_3$.

applied to explore the physicochemical properties of the photoanode after long-term testing. The SEM image shows the morphology of $SnO_{2-x}/BiVO_4/WO_3$ is well-maintained (Supplementary Fig. 31a) and a conformally and stable $SnO_{2-x}$ overlayer on $BiVO_4$ is demonstrated (Supplementary Fig. 31b). Furthermore, EPR measurement evidence the maintenance of the oxygen vacancies of the $SnO_{2-x}$ layer (Supplementary Fig. 31c). Moreover, XPS investigations on the composition of the surface and interfaces of $SnO_{2-x}/BiVO_4/WO_3$ were carried out. As shown in Supplementary Fig. 32a, the O1s peak shows no detectable changes in intensity and position, whereas a slight shift of the V 2p towards lower binding energy and reduced intensity can be observed after long-term testing. The phenomenon can be tracked to the photo-charging of $BiVO_4$ by which the $V^{5+}$ is somehow reduced to $V^{4+}$. Nevertheless, the Sn 3d peaks are determined to be stable during long-term testing (Supplementary Fig. 32b). On the other hand, the Mo-SACs/mrG-GDE also shows good stability with well-maintained singly dispersed atoms (Supplementary Fig. 33a) and quasi-nanoarray aerophilicity areas (Supplementary Fig. 33b). The modified oxygen-diffusion-benefit structure is supposed to attribute to good stability

since the insufficient oxygen supplement during large current density would damage the catalysts. Overall, it is assumed that the optimized structure attributes to the good stability of the device.

Further on, detailed and systematic studies have been performed to evaluate the ROSs generation/interconversion as well as the reaction mechanism in our self-cycled Fenton-like system. First, H-cell configuration was taken to specifically investigate the roles of $^1O_2$ and $\cdot O_2^-$ during the $H_2O_2$ production in both cathode and anode since both of them are common intermediates during the photocatalytic ORR for $H_2O_2$ production[51]. For the cathode part, the scavenger experiments using p-benzoquinone (BQ, a sacrificial agent for $\cdot O_2^-$) and 2, 2, 6, 6-tetramethylpiperidine (TEMP, a sacrificial agent for $^1O_2$) over PTFE@Mo-SACs/mrG-GDE were performed. As shown in Supplementary Fig. 34a, b, the amount of the $H_2O_2$ is invariant with respect to the concentration of both BQ and TEMP, indicating that the quenching of either $^1O_2$ or $\cdot O_2^-$ has no effect on the $H_2O_2$ formation during the ORR in the cathode part[52]. Similar experiment results were also observed in the anode part (Supplementary Fig. 34c, d) that neither the quenching of $^1O_2$ nor $\cdot O_2^-$ has impact on the generation of $H_2O_2$ during the WOR in

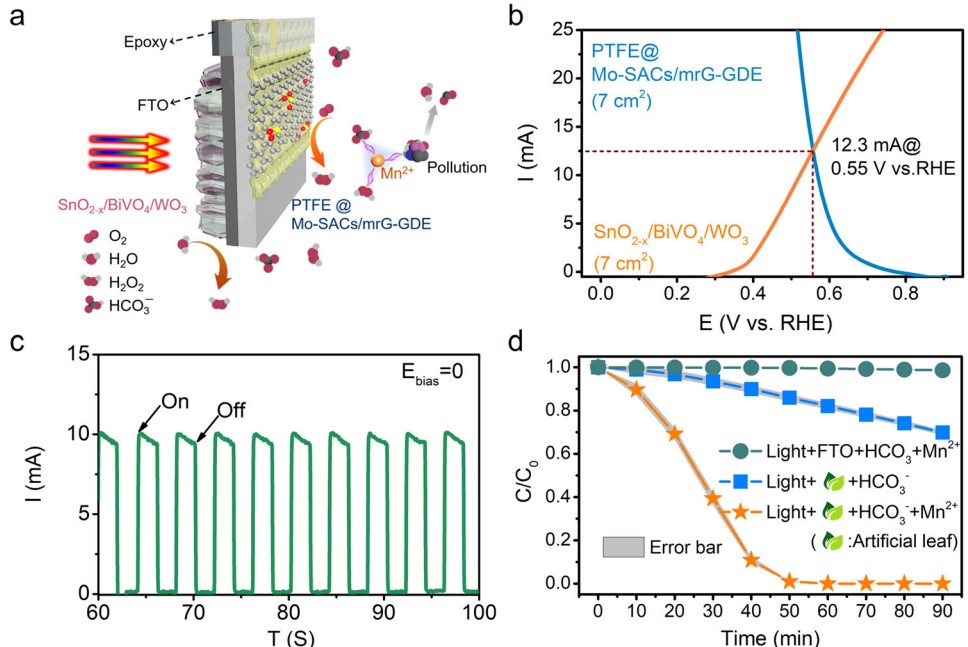

**Fig. 6 | Design of a large artificial leaf for PEC-driven wastewater treatment.**
**a** Schematic of the artificial leaf for water treatment. **b** Coupled LSV scan of the
$SnO_{2-x}/BiVO_4/WO_3$ photoanode (7 cm²) and PTFE@Mo-SACs/mrG-GDE cathode
(7 cm²). **c** Photocurrent−time profiles of $SnO_{2-x}/BiVO_4/WO_3$‖PTFE@Mo-SACs/mrG-
GDE under bias-free conditions. **d** Degradation of 15 ppm NP with consumption of
only oxygen and sunlight. Reaction electrolyte: 15 mL of 0.4 M $NaHCO_3$ solution.

the anode part. Moreover, RRDE tests have shown that the main pro-
duct of $O_2$ reduction in the cathode is $H_2O_2$ (Fig. 4a). Above all, we
believed that the $^1O_2$ and $\cdot O_2^-$ are converted from the in situ generated
$H_2O_2$ in the self-cycled photo-like system.

Next, the sacrificial agent experiments were conducted to inves-
tigate the main ROSs in the self-cycled photo-Fenton-like system with
and without $Mn^{2+}$ species. As shown in Supplementary Fig. 35, sacrifi-
cial agents, including tert-butanol (TBA, a sacrificial agent for $\cdot OH$), BQ,
and TEMP have been applied for the investigation[53]. In the system
without $Mn^{2+}$ species, a barely decrease in the degradation rate (from
30.2 to 29.5% at 90 min) is observed when TEMP is added, while a small
inhibitory effect on the degradation (from 30.2 to 27.6% at 90 min) is
exhibited in the presence of BQ, suggesting the more positive role of
$\cdot O_2^-$ than $^1O_2$ in the degradation process (Supplementary Fig. 35a).
Specifically, the removal effect is almost inhibited after the addition of
TBA, suggesting two reasons: 1. the $\cdot OH$ is the main ROS during the
degradation. 2. the $\cdot O_2^-$ and $^1O_2$ might come from the $\cdot OH$ in the system
without the $Mn^{2+}$ species. On the other hand, the addition of BQ, TEMP,
and TBA leads to a certain decrease in the degradation rate in the
system containing $Mn^{2+}$ species (decreased from 90.5% to 80.7%, 71.5%,
and 28.1% at 40 min for the addition of TEMP, BQ and TBA, respec-
tively) (Supplementary Fig. 35b). Notably, an effective removal rate
after the addition of TBA is still observed in the presence of $Mn^{2+}$
species (28.1% at 40 min) compared to that without $Mn^{2+}$ species (0.2%
at 40 min), indicating that the different interconversion mechanisms
of ROSs between the system with and without $Mn^{2+}$ species, which will
be further evaluated below.

To further investigate the role $HCO_3^-$ in the generation of ROSs of
the self-cycled photo-like system, KPi electrolyte with the same pH
value (adjusted by KOH) is taken to replace $NaHCO_3$ electrolyte. As
shown in Supplementary Fig. 36a, barely degradation effect is
observed in the absence of $Mn^{2+}$, even if $H_2O_2$ is effectively generated
in the reaction system (via 2e⁻ORR in cathode) (Supplementary
Fig. 36b), which indicates KPi cannot activate $H_2O_2$ into ROSs as the
$HCO_3^-$ does (Fig. 6d and Supplementary Fig. S1). Furthermore, the
degradation rate in KPi electrolyte slightly raised to 5.4% at 40 min
when $Mn^{2+}$ is added, which is still much lower than that in $HCO_3^-$

electrolyte with $Mn^{2+}$ (89.5% at 40 min). That means the effective
activation of $H_2O_2$ is largely suppressed in the absence of $HCO_3^-$. EPR
technology was employed to further investigate the mechanism.
Methanol and DMPO were added as capture agents to exclude the
interface of $\cdot OH$ for detecting $\cdot O_2H$[54]. As shown in Supplementary
Fig. 37a, no signals of $\cdot O_2H$ are detected in the system, indicating $\cdot O_2H$
not being generated. The signal intensities of $\cdot O_2^-$ are unchanged after
the addition of TEMP while being disappeared after the addition of
TBA, indicating that the $\cdot OH$ is essential for the generation of $\cdot O_2^-$ while
$^1O_2$ is not[55](Supplementary Fig. 37b). Besides, no obvious signal is
observed in the detection of $^1O_2$ (Supplementary Fig. 37c). Moreover,
the addition of TEMP and BQ has no impact on $\cdot OH$ (Supplementary
Fig. 37d). Therefore, the mechanism for the ROSs generation could be
given:

$$HCO_3^- + H_2O_2 \leftrightarrow HCO_4^- + H_2O \quad (4)$$

$$HCO_4^- + H_2O_2 \rightarrow HCO_3^- + \cdot OH + HO_2^- \quad (5)$$

$$\cdot OH + HO_2^- \rightarrow \cdot O_2^- + H_2O \quad (6)$$

$$\cdot OH + \cdot O_2^- \rightarrow {}^1O_2 + OH^- \quad (7)$$

On the other hand, the addition of $Mn^{2+}$ in $HCO_3^-$ electrolyte could
largely increase the degradation rate by improving the generation of
both $^1O_2$, $\cdot O_2^-$ and $\cdot OH$ (Fig. 6d and Supplementary Fig. S1). While the
$Mn^{2+}$ in KPi electrolyte shows a little effect of degradation (Supple-
mentary Fig. 36a), indicating that the complex between $Mn^{2+}$ and
$HCO_3^-$ plays the main role in the generation of the ROSs, which is also
proved by the Fig. S1. EPR measurements were performed to further
investigate the mechanism. Unlike in the $HCO_3^-$ system, the addition of
TBA has little impact on the generation of $\cdot O_2^-$ and $^1O_2$ during the EPR
measurements, indicating that $\cdot OH$ is not essential for generating them
in the $Mn^{2+}(HCO_3^-)_n$ system (Supplementary Fig. 38a, b). However, the
signal for $^1O_2$ disappeared after the addition of BQ, indicating $^1O_2$ come

from $\cdot O_2^-$ in the $Mn^{2+}(HCO_3^-)_n$ system (Supplementary Fig. 38a). Moreover, the addition of TEMP and BQ has no impact on $\cdot OH$ (Supplementary Fig. 38c). Above all, the ROSs generation in the $Mn^{2+}$–$HCO_3^-$ system is given:

$$Mn^{2+}(HCO_3)_n + H_2O_2 \rightarrow Mn^{4+}(HCO_3)_m + \cdot OH \qquad (8)$$

$$Mn^{4+}(HCO_3)_m + H_2O_2 \rightarrow \cdot O_2^- + Mn^{2+}(HCO_3)_n \qquad (9)$$

$$Mn^{4+}(HCO_3)_m + \cdot O_2^- \rightarrow {}^1O_2^- + Mn^{2+}(HCO_3)_n \qquad (10)$$

In the self-cycled system, the transformation of $Mn^{IV}$ species into $Mn^{II}$ species would be largely enhanced through Eq. (11) by the reduction of a cathode according to Fig. 1b and Supplementary Fig. S3.

$$Mn^{4+}(HCO_3)_m + e^-_{(cathode)} \rightarrow Mn^{2+}(HCO_3)_n \qquad (11)$$

## Discussion

In summary, we report an unassisted solar-driven self-cycled photo-Fenton-like system through in situ production and utilization of $H_2O_2$ using an artificial leaf. First, we confirm that the Mn(II)/(IV) redox can be effectively recycled in the PEC system with the production of various ROSs, including $\cdot OH$, $O_2^{\cdot-}$ and $^1O_2$ driven by $H_2O_2$ activation and photoinduced cathode reduction under bicarbonate media. Next, in order to fulfill the adequate $H_2O_2$ supplement, comprehensive guidance toward enhancing PEC-to-$H_2O_2$ efficiency was emphasized with kinetic optimization of both the photoanode and cathode. Dramatically improved stability and current density due to the enhanced carrier transport were fulfilled in the photoanode, and an interfacial strategy was proposed to construct an oxygen local confinement structure to simultaneously benefit both the electron-transfer and mass-transfer processes at the cathode. As a result, the high recorded unassisted $H_2O_2$ production rate of 0.77 μmol/(min cm$^2$) and an SHyE of 1.46% were achieved by coupling the optimized photoanode and cathode. Especially, a 70-cm$^2$ artificial leaf with 30 days of stability was successfully realized as an effective self-cycled photo-Fenton-like system for wastewater treatment with the consumption of only oxygen, water, and sunlight. Notably, some improvements in this PEC-driven wastewater treatment system are needed for industrial-scale applications, including enhanced solar utilization efficiency, continuous oxygen supplement, and tuning electrolyte components. Therefore, additional discussion for addressing the above issues was carried out to identify the limitations, which are expected to be addressed to achieve further breakthroughs (see Supplementray discussion). Overall, the design of effective solar-fuel device presented here is expected to inspire further sustainable application, while the successful demonstration of a one-pot self-cycled wastewater treatment system highlights the promising potential of artificial photosynthesis technology in decentralized environmental governance applications.

## Methods

### Synthesis of SnO$_{2-x}$/BiVO$_4$/WO$_3$

The WO$_3$ nanosheet array used as the template electrode was first synthesized by a modified hydrothermal method. Briefly, a precursor solution was prepared by slowly dropping 2 mL of 3.5 M HCl into 25 mL of 25 mM sodium tungstate dihydrate aqueous solution and allowing the reaction to occur for 1 min. Then, 25 mM ammonium oxalate was dissolved in the precursor solution and reacted for 15 min until the solution became completely transparent and colorless. Stirring was performed throughout the whole dissolution and reaction process. Then, a bare fluorine-doped tin oxide (FTO) substrates (2 × 2 cm$^2$) was placed against the Teflon wall before the solution was poured, and the hydrothermal reaction was carried out in a sealed 40 mL Teflon

autoclave at 110 °C for 60 min. The obtained WO$_3 \cdot (H_2O)_x$ nanoarray was washed with deionized (DI) water, dried, annealed at 500 °C in the air for 3 h, and cooled naturally to obtain the WO$_3$ nanoarray. Next, the BiVO$_4$ precursor solution was synthesized by dissolving 100 mM bismuth nitrate pentahydrate and 110 mM ammonium metavanadate in 20 mL of ethylene glycol. After that, a certain amount of BiVO$_4$ precursor solution was dropped onto the WO$_3$ photoanode (10 μL for 1 cm$^2$), followed by annealing at 500 °C in the air for 2 h and cooling naturally. Then, the obtained sample was placed into a 0.1 M NaOH solution to remove excessive vanadium oxide and obtain the BiVO$_4$/WO$_3$ photoanode. Finally, the SnO$_{2-x}$ layer was deposited according to our previous work to obtain the SnO$_{2-x}$/BiVO$_4$/WO$_3$ photoanode[27].

### Synthesis of Mo-SACs/mrG and mrG

Graphene oxide (GO) was synthesized by oxidation of natural graphite flakes according to a modified Hummers' method. Mo-SACs/mrG was synthesized using a one-step process. In a specific procedure for synthesizing Mo-SACs/mrG, a pristine GO suspension was washed with DMSO several times to obtain a GO/DMSO solution with a GO concentration of 0.5 mg mL$^{-1}$. Then, 250 μL of 3 mg mL$^{-1}$ MoCl$_5$/DMSO was added to 30 mL of a 0.5 mg mL$^{-1}$ GO/DMSO solution. The homogeneous solution was transferred into a 50 mL Teflon-lined autoclave, which was sealed and maintained at 135 °C for 12 h in an oil bath. Finally, the resulting Mo-SACs/mrG was freeze-dried to obtain a powder after removing the DMSO solvent. The mrG was synthesized as same as Mo-SACs/mrG but without adding metal elements.

### Synthesis of PTFE@Mo-SACs/mrG-GDE

The Mo-SACs/mrG ink was prepared by dissolving Mo-SACs/mrG powder in DI-water/ethanol/5 wt.% Nafion solution (volume ratio of 9:1:0.05) at a concentration of 1 mg/mL and stirring for 48 h. Then, the catalyst ink was obtained by adding a certain amount of poly tetrafluoroethylene (PTFE) aqueous solution (0.1 wt.%) to the Mo-SACs/mrG ink and stirring for another 48 h to obtain Mo-SACs/mrG nanosheets with evenly distributed PTFE nanoparticles at the required mass ratio (PTFE@Mo-SACs/mrG). After that, the catalyst ink was dropped on the Teflon-treated hydrophobic carbon fiber (TCF) with a loading amount of 0.5 mg/cm$^2$, followed by natural drying to obtain PTFE@Mo-SACs/mrG-GDE.

### Synthesis of Mo-SACs/mrG-GDL electrode

Mo-SACs/mrG-GDL electrode was prepared by dropping Mo-SACs/mrG ink on the gas diffusion layer coated carbon fiber with a loading amount of 0.5 mg/cm$^2$.

### Synthesis of PTFE@mrG-GDE

PTFE@mrG ink was prepared the same as the PTFE@Mo-SACs/mrG-GDE except for changing the Mo-SACs/mrG into mrG.

### Synthesis of the artificial leaf

For the wireless artificial leaf structure, an Ag paste-Cu sheet-Ag paste was used to connect the SnO$_{2-x}$/BiVO$_4$/WO$_3$ photoanode to the PTFE@Mo-SACs/mrG-GDE cathode. Whole ohmic contacts were covered with epoxy for waterproofing.

### Material characterizations

Scanning electron microscope (SEM) images were recorded on a field-emission scanning electron microscope (FESEM, JSM-7800F, Japan). X-ray diffraction (XRD) patterns were obtained with a D500/5000 diffractometer in Bragg-Brentano geometry under Cu Kα radiation. High resolution transmission electron microscopy (HR-TEM) images were collected on a FEI Talos F200X JEOL electron microscope with energy-dispersive X-ray spectroscopy (EDS). High-angle annual dark field-scan transmission electron microscopy (HAADF-STEM) was performed on a JEOL JEM-ARM200F TEM/STEM (200 kV) with a spherical

aberration corrector. The X-ray photoelectron spectroscopy (XPS) measurements were performed on a PHI-5000versaprobeIII. The spectra were calibrated by the reference of the C1s peak at 284.8 eV. Ultraviolet-visible (UV-vis) diffuse reflectance spectra were obtained via a UV−vis spectrometer (Shimadzu UV-3600 spectrophotometer) equipped with an integrating sphere. Electron paramagnetic resonance (EPR) measurements was carrier out by a Bruker EMX-plus spectrometer. The microwave power was 6.325 mW, microwave frequency was 9.826 GHz, and temperature was 298 K.

## PEC measurements

There are two systems (a three-electrode system and a two-electrode system) for the photoelectrochemical (PEC) measurements to investigate the activity of photoanode/tandem device. Specifically, the three-electrode system was carried out using a photoanode as a working electrode, saturated Ag/AgCl as a reference electrode, and a carbon electrode or designed cathode as a counter electrode, among which the designed cathode was taken as a counter electrode only in the tandem device. The two-electrode system was carried out to evaluate the tandem device using a photoanode as a working electrode and a cathode as a counter electrode. All potentials were converted to the reversible hydrogen electrode (RHE) reference scale by the equation E (V vs. RHE) = E (V vs. Ag/AgCl) + 0.0591 × pH + 0.197. The illumination source was a 300 W Xe arc lamp with an AM 1.5G filter (100mW/cm2, FX 300, Beijing PerfectLight Co. Ltd). Electrochemical Impedance Spectroscopy (EIS) measurements were carried out with a AC perturbation of 10 mV applied over the frequency range of $10^{-2}$ Hz-$10^6$ Hz.

## ORR measurements

Electrochemical oxygen reduction reaction (ORR) tests were performed in a CS2350H potentiostat (Corrtest, Wuhan) with a three-electrode cell at room temperature. A RRDE assembly (AFE7R9GCPT, Pine Instruments) consisting of a glassy carbon rotating disc electrode ($\Phi$ = 5.0 mm) and a Pt ring ($\Phi$ = 8.0 mm) was used, with a theoretical collection efficiency of 37%. A glassy carbon electrode loaded with catalyst was used as the working electrode. The working electrodes were prepared by dispersing the catalyst powder in ethanol and 5% Nafion 117 solution (20 μL of Nafion for 1 mL of ethanol) to achieve a catalyst concentration of ~1 mg mL$^{-1}$. After sonication for 60 min, 7 μL of the catalyst ink was drop-dried onto a glassy carbon disc (area: 0.196 cm$^2$). The ORR activity and selectivity were investigated from polarization curves and RRDE measurements conducted in an oxygen-saturated electrolyte. The H$_2$O$_2$ selectivity was calculated using the following equation: $\eta(\%) = 200 \times (I_{Ring}/N)/(I_{Disc} + I_{Ring}/N)$. The electron-transfer number ($n$) at the disc electrode during the ORR was calculated using $n = 4 - (2 \times \eta_{H2O2})$, where $I_{Ring}$ is the ring current, $I_{Disc}$ is the disc current and $N$ is the collection efficiency.

## H$_2$O$_2$ production efficiency evaluation and organic dye degradation measurements

A H-type quartz reactor with two-compartment was used to evaluate both the anodic and cathodic H$_2$O$_2$ production efficiencies of the artificial leaf. The two-compartment reactor contained a 0.4 M NaHCO$_3$ solution in both the photoanode and cathode chambers separated by Nafion 117 membrane. The electrolyte solution in cathode chambers was first purged with pure O$_2$ for 30 min before testing, and the pure O$_2$ was continuously fed for stability testing. All current densities were normalized to the geometrical area of the photoanode. An integrated PEC system (PEC1000, Beijing Perfectlight Technology Co., Ltd.) with a solar simulator (AM 1.5G) was used.

The artificial leaf was placed in a single-compartment quartz reactor under AM 1.5 illumination with continuous O$_2$ purging to carry out the dye degradation tests, the electrolyte was 15 mL of 0.4 M NaHCO$_3$ solution containing 15 ppm NP and 0.04 ppm Mn(II) when the active area of the artificial leaf was 7 cm$^2$. Furthermore, the wastewater

was changed to 200 mL of 0.4 M NaHCO$_3$ solution containing 5 ppm rhodamine B (Rh. B), 5 ppm methylene blue (MB), 5 ppm NP and 0.04 ppm Mn(II) when the active area of the artificial leaf was extended to 70 cm$^2$. Sample aliquots was taken at regular intervals and tested quickly. The concentration of 4-nitrophenol (NP), methylene blue (MB) and rhodamine B (Rh. B) wereas measured by a Ultraviolet-visible (UV-vis) spectremeter (PE lambda 750 spectrophotometer).

## Quantification of H$_2$O$_2$

H$_2$O$_2$ evolution was detected using the *N,N*-diethyl-1,4-phenylene-diamine (DPD) method. A stock solution of DPD was prepared by dissolving 0.1 g of DPD in 10 mL of a 0.05 M H$_2$SO$_4$ solution. A peroxidase (POD) solution was prepared by dissolving 10 mg of POD in 10 mL of deionized water and was kept in a refrigerator until use. A potassium phosphate buffer solution was prepared by mixing 49.85 mL of deionized water, 43.85 mL of 1 M monobasic potassium phosphate, and 6.3 mL of 1 M potassium phosphate. Sample aliquots (2 mL) were collected by a syringe during irradiation and mixed with 0.4 mL of potassium phosphate buffer solution, 3 mL of water, 0.05 mL of DPD, and 0.05 mL of POD and were shaken for 90 s. The obtained solutions were analyzed by Ultraviolet-visible (UV−vis) spectremeter (PE lambda 750 spectrophotometer).

## Radical quenching tests

The quenching tests for the identification of reactive oxygen species (ROSs) were carried out based on the EPR technology with different sacrificial agents:p-benzoquinone (BQ, a sacrificial agent for ·O$_2^-$), 2, 2, 6,6-tetramethylpiperidine (TEMP, a sacrificial agent for $^1$O$_2$) and tert-butanol (TBA, a sacrificial agent for ·OH). The sacrifical agents were added before the reaction began and other steps were the same with experiments above.

## Data availability

The data described in this paper are available from the authors upon reasonable request.

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

## Acknowledgements

This work was supported by NSFC (22172077, 21902104) and the International Cooperation Program (BZ2020063) of Jiangsu Province, the Natural Science Foundation of Jiangsu Province of China (BK 20211573), the Fundamental Research Funds for the Central Universities (30921011216). This research was also supported by the Yonsei Signature Research Cluster Program of 2021 (2021-22-0002). J.H. Park acknowledges the support from NRF Korea (2019R1A2C3010479, 2021M3H4A1A03049662, 2022H1D3A3A01077254, 2022R1A4A200823).

## Author contributions

K.Z. and J.H.P. conceived and designed the experiments. Y.H.A. and J.Y.S. conducted the dye degradation measurement and analysis, C.R.D. and L.Y.W. carried out materials synthesis and electrochemical characterization. Y.L.Y. and X.M.H. participated in part of the synthesis. Y.C. and G.J. carried out materials characterization. K.Z., C.R.D., Y.H.A., and J.H.P. co-wrote the paper. All authors discussed the results and commented on the manuscript.

## Competing interests

The authors declare no competing interests.

## Additional information

**Supplementary information** The online version contains

supplementary material available at https://doi.org/10.1038/s41467-022-32410-0.

