## [Peer Review File · Nature Communications]

REVIEWER COMMENTS

Reviewer #1 (Remarks to the Author):

In this research work, the authors have constructed a self-cyclable photo-Fenton-like system with an artificial leaf for the production of H₂O₂ from water. The system was also scaled up to 70 cm² to be utilized in an unassisted solar-driven bicarbonate-activated hydrogen peroxide system, which is inspiring for future breakthroughs for sustainable environmental applications. Although the authors have conducted the experiment with excellent care, there is a lack of fundamental knowledge and insights via computational simulations. It is crucial to integrate experiment and computation for aiming for a top-tier journal to reflect the nature of fundamental and application-driven processes. In view of the extremely high requirement, impacts, and novelty of Nature Communications, this work should be rejected without further review. This work can be considered in the sister journal (e.g. Communications Chemistry) after appropriate revisions from the authors.

- (1) The authors should link how their PEC system correlates to an artificial leaf.
- (2) Include a brief description of the limitations to be addressed in the conclusion. The authors can also mention that a more detailed discussion can be found in the supplementary information.
- (3) The methodology should be more detailed. Provide details on the reaction medium used for water and bicarbonate-activated H₂O₂ production, as well as the reactor. How long is O₂ purged through the electrolyte? The present methodology is not well described and sound.
- (4) The first-principles DFT calculations are necessary to bridge experiment and calculation. This is vital to be included in the manuscript to unravel the reaction pathway. The density of states (DOS) should also be studied.
- (5) It is important to include a video on the reactor setup for the solar-to-H₂O₂ conversion for both lab-scale and prototype devices.
- (6) The authors should provide a comparison table to compare the present results with the state-of-the-art catalytic performance of H₂O₂ production. This is extremely important to advance state-of-the-art research in the realm of renewable H₂O₂ production.
- (7) The authors observed no deterioration after one month showing the robustness of the artificial leaf. It is important to further understand and comprehensively characterize the physicochemical properties of the spent catalysts.
- (8) Minor: Carefully check the entire manuscript for grammatical errors. For example, in line 63-65, "In recent years, bicarbonate (HCO₃⁻) was found that not only can react with H₂O₂ to form HCO₄⁻ with high reactivity, but also can act as an auxiliary for water oxidative H₂O₂ evolution under photoelectrochemical (PEC) condition..." , and many more.

Reviewer #2 (Remarks to the Author):

This manuscript reported unassisted solar H₂O₂ production by combining SnO₂/BiVO₄/WO₃ || PTFE@Mo-SAC/mrG-GDE in a bicarbonate-containing electrolyte with a high production rate of 0.77 μmol/min cm⁻² and achieved efficiency of 1.46%. This combined system showed 150 % faradaic efficiency from both the photoanode and cathode. When demonstrated the large-scale artificial leaf, unassisted current of 10.7 mA with a production rate of 5.17 μmol min⁻¹ is achieved by this system. Furthermore, the successful application was found when such a system was applied for the degradation of 4-nitrophenol in one-pot self-cycled wastewater treatment. Overall, the manuscript introduces a novel concept for the dual application of artificial photosynthesis of H₂O₂ and greener wastewater treatment. The manuscript is a well-written interesting paper and is publishable in “nature communications” after the modifications considering comments below.

1. In the energy level diagram, in Fig. 2, the authors should mention the reference if the values of energy are not measured by themselves or mention the name of equipment used for measuring the values.

2. In Fig. 5a, the intersection point of both the electrodes exhibited 2.07 mA cm⁻², whereas the current was reduced to 1.56 mA cm⁻² in Fig. 5b. The authors need to explain this difference and need to measure LSV at lower scan rates too to exactly determine the intersection current.

3. The authors need to add the error bars in Fig. 3g, Fig. 4b, Fig. 4e, Fig. 5d, and Fig. 6d to increase the reliability of data.

4. Some minor errors. This includes:

- Fig. 5e cm² on the Y1 axis, Fig. 6 legends 7 cm² is indicated instead of 70 cm² and also, in Fig. 6b.
- Citation of Fig. in line 88.
- Line 140, the subscript of production rate.
- Numerical value of H₂O₂ selectivity in line 222.

- mv/dec to mV/dec.
- On page 8, line 157, there is no end bracket next to 'Supplementary Fig. 6'
- From method section, line 399, authors synthesized WO₃ at 110°C for 40 min, but from Table S1, optimized WO₃ is synthesized at 110°C for 60 min. Make them clear.
- From Fig. 5c, 5d and 6c, change time unit from capital 'S' to small 's'

Reviewer #3 (Remarks to the Author):

In this manuscript, the author designs a self-cycled photo-Fenton-like system for real-time wastewater purification. This is an interesting study. However, in terms of the mechanistic discussion, the authors do not consider the role of reactive species other than hydroxyl radicals at all. Therefore I don't think it can be published.

1. It is well known that the removal of pollutants may be the result of the combined action of multiple active species. However, only the role of hydroxyl radicals is explored in the manuscript, and the possible roles of other reactive species such as 1O_2 and O_2^- are not considered..
2. As mentioned before, the authors constructed a relatively complex system including catalysts, O_2 , HCO_3^- , Mn(II) and Mn(III), etc. For such a system, the contribution of multiple active species to pollutant degradation and the possible mutual transformation between them should be systematically considered. In fact, there may be 1O_2 and O_2^- intermediates for the formation of H_2O_2 . Therefore, I do not think it is reliable to attribute removal of pollutants entirely to hydroxyl radical.
3. The author proposes that the generation process of H_2O_2 is $O_2 + 2H_2O \rightarrow 2H_2O_2$. If the external energy output is sufficient to drive this reaction, holes may directly oxidize H_2O/OH^- to $\bullet OH$. Did the author consider this process?

Response to the Comments on Nature Communications

Manuscript ID: NCOMMS-22-52128

Dear Reviewers:

We would like to thank the reviewers for their careful reading and helpful comments. We revised the manuscript thoroughly according to the comments. The added items are highlighted in red in the main manuscript and the supplementary information. Following changes were made and listed below:

Reviewer(s)' Comments to Author:

Reviewer #1: In this research work, the authors have constructed a self-cyclable photo-Fenton-like system with an artificial leaf for the production of H₂O₂ from water. The system was also scaled up to 70 cm² to be utilized in an unassisted solar-driven bicarbonate-activated hydrogen peroxide system, which is inspiring for future breakthroughs for sustainable environmental applications. Although the authors have conducted the experiment with excellent care, there is a lack of fundamental knowledge and insights via computational simulations. It is crucial to integrate experiment and computation for aiming for a top-tier journal to reflect the nature of fundamental and application-driven processes. In view of the extremely high requirement, impacts, and novelty of Nature Communications, this work should be rejected without further review. This work can be considered in the sister journal (e.g. Communications Chemistry) after appropriate revisions from the authors.

Our Response: We thank you very much for your evaluation of our work. The questions and suggestions raised by you are extremely important and helpful, which make us deep thought, thereby improving the quality of our work in the revision. As the reviewer said, the main target of this work is to construct a self-cycled photo-Fenton-like system with high solar-to-H₂O₂ (STH) efficiency. The kinetic optimization of the photoanode and cathode is the key to achieve a high STH efficiency towards pollutant degradation application, and the modified approaches are extended researches based on our previous work which carefully explored the activity of the catalyst with systematically theoretical simulation. The related DFT calculation will be introduced in the following parts and we have done a series of investigations on morphology, chemical composition, valance state, catalysis performance et.al. The presentation video is also newly included. The main contribution of our work is further strengthened, and we believe the quality of the paper is significantly improved.

#1: The authors should link how their PEC system correlates to an artificial leaf.
--

Our Response: Thanks for your wise suggestion. In our work, the PEC measurements in a three-electrode system are used to demonstrate the operating possibility of the artificial

leaf as well as achievable theoretical current density (efficiency). For actual H₂O₂ measurements, a two-electrode PEC system was conducted without using applied bias (Fig. 6c in the manuscript). The operating conditions in the two-electrode PEC system are the same as that in the artificial leaf, as both are light-driven reactions. We have already claimed this point in our manuscript.

1#2: Include a brief description of the limitations to be addressed in the conclusion. The authors can also mention that **a more detailed** discussion can be found in the supplementary information.

Our Response: Thanks for your kind advice. We have briefly added the description about limitations in the conclusion, as follows:

“Notably, some improvements in this PEC-driven wastewater treatment system are needed for industrial-scale applications, including the enhanced solar utilization efficiency, continuous oxygen supplement, and tuning electrolyte components. Therefore, additional discussion for addressing the above issues was carried out to identify the limitations, which are expected to be addressed to achieve further breakthroughs.”

and more detailed discussions have been added in our supplementary information as follows:

1. **The solar utilization efficiency.** One of the significant goals of our system is to build a sustainable AOPs system with self-generated H₂O₂ based on the PEC technology. A Higher H₂O₂ generation rate is highly expected to accelerate the rate of AOPs due to the rapid consumption of H₂O₂. Based on that concept, kinetic optimization of photoanode and cathode is carefully and deliberately investigated to raise the H₂O₂ generation rate, namely, STH efficiency, as much as possible. Even though, we expect that the STH efficiency can be substantially raised by introducing the state-of-art photovoltaic (PV) in the PEC system because the only part range of solar energy is absorbed (<510 nm) due to the relatively large band gap of the SnO_{2-x}/BiVO₄/WO₃ photoanode (~2.45 eV). Therefore, the combination of a PV cell with a complementary absorption range (e.g. ~1.2 eV) could fulfill a broader absorption range with an extra driving force, which could lead to a higher working current density of the tandem PEC system and the artificial leaf.

According to previous theoretical research (Ref. R1), the unassisted working current density is expected to be raised by 3-4 times when a qualified perovskite cell was combined.

2. Supplement of oxygen. In this work, the oxygen supplement for the 2e-ORR in the cathode relies on the aeration from a high-pressure oxygen bottle during the whole reaction process. Therefore, it would make large progress if the oxygen/air could naturally diffuse into the cathode without external compression. Recently, a few researchers reported the concept of a deliberately designed ORR cathode with a superhydrophobic interface that allows the rapid and natural transport of O₂ from the atmosphere to the cathode through the backside of the electrode (Ref. R2, 3). Our work, on the other hand, takes care of both conductivity and oxygen diffusion efficiency by creating quasi-nanoarray aerophilicity areas only on the electrocatalysts coating side (front side). Therefore, it is reasonable to believe that an “asymmetric sandwich” electrode with modified electrocatalysts coating the front side and oxygen natural diffusion backside could avoid aeration energy consumption with improved electrocatalysis performance at the same time, which is believed to substantially enhance the feasibility of our device application.

3. Electrolyte in the natural environment. The environmental benignity is one of the most prospecting merits of our device because the reactors and the products involved are only oxygen, water and hydrogen peroxide and the electrolyte is bicarbonate aqueous solution, which is ubiquitous in groundwater, showing strong competitiveness sustainably compared to the traditional AOPs. However, the relatively high electrolyte concentration (0.4 M HCO₃⁻) is necessary for high H₂O₂ generation current density in the PEC system. Therefore, we expect a potential electrolyte with a lower concentration for high H₂O₂ generation current density in further research.

We hope that this discussion of the limitation and potential approaches will inspire further research toward further exploration of effective and sustainable wastewater treatment as well as efficient solar-fuel production/utilization devices. This part is included in the supplementary discussion.

1#3: The methodology should be more detailed. Provide details on the reaction medium used for water and bicarbonate-activated H₂O₂ production, as well as the reactor. How long is O₂ purged through the electrolyte? The present methodology is not well described and sound.

Our Response: Thanks for your kind advice. We are sorry for the unclear description of the methodology in the previous version. We have detailedly described our methodology in our revised manuscript, which is believed to be easily followed by interested readers.

PEC Measurements. There are two systems (three-electrode system and two-electrode system) for the PEC measurements to investigate the activity of photoanode. Specifically, the three-electrode system was carried out using photoanode as working electrode, saturated Ag/AgCl as reference electrode and carbon electrode or designed cathode as counter electrode, among which the designed cathode was taken as counter electrode only in the tandem device. The two-electrode system was carried out to evaluate the tandem device using photoanode as working electrode and cathode as counter electrode. All potentials were converted to the RHE reference scale by the equation $E \text{ (V vs. RHE)} = E \text{ (V vs. Ag/AgCl)} + 0.0591 \times \text{pH} + 0.197$.

H₂O₂ production efficiency evaluation and organic dye degradation. A H-type quartz reactor with two-compartment was used to evaluate both the anodic and cathodic H₂O₂ production efficiencies of the artificial leaf. The two-compartment reactor contained a 0.4 M NaHCO₃ solution in both the photoanode and cathode chambers separated by Nafion 117 membrane. The electrolyte solution in cathode chambers was first purged with pure O₂ for 30 min before testing, and the pure O₂ was continuously fed for stability testing. All current densities were normalized to the geometrical area of the photoanode. An

integrated PEC system (PEC1000, Beijing Perfectlight Technology Co., Ltd.) with a solar simulator (AM 1.5G) was used.

The artificial leaf was placed in a single-compartment quartz reactor under AM 1.5 illumination with continuous O₂ purging, the electrolyte was 15 mL of 0.4 M NaHCO₃ solution containing 15 ppm NP and 0.04 ppm Mn(II) when the active area of the artificial leaf was 7 cm². Furthermore, the wastewater was changed to 200 mL of 0.4 M NaHCO₃ solution containing 5 ppm rhodamine B (Rh. B), 5 ppm methylene blue (MB), 5 ppm NP and 0.04 ppm Mn(II) when the active area of the artificial leaf was extended to 70 cm².

1#4: The first-principles DFT calculations are necessary to bridge experiment and calculation. This is vital to be included in the manuscript to unravel the reaction pathway. The density of states (DOS) should also be studied.

Our Response: Thanks for your kind advice. We agree that the combination of theoretical calculations with experiments is essential to understanding the mechanism of H₂O₂ generation. Actually, for both anodic and cathodic H₂O₂ evolutions, the used materials were elaborately screened based on our previous works (Ref. R4 and R5).

For the anode part, the water oxidative H₂O₂ evolution by SnO_{2-x}/BiVO₄ was investigated by both theoretical calculations with experimental measurements, where both 2e⁻ reaction and 1e⁻ reaction pathways were confirmed. In detail, for the BiVO₄ photoanode, the reaction step from absorbed OH* to H₂O₂ evolution is downhill in the free energy, suggesting much more favorable thermodynamics for H₂O₂ evolution while the SnO_{2-x} exhibits thermodynamically unfavorable H₂O₂ evolution as a result of a much higher free energy barrier for H₂O₂ formation. It turns out that the pinned quasi-Fermi level on SnO_{2-x}/BiVO₄ surface caused by the thin SnO_{2-x} coating layer facilitates the kinetically favored surface hole transfer (from four-electron water oxidation reaction (4e-WOR) and 2e-WOR competitive reactions into one-electron water oxidation reaction (1e-WOR) and 2e-WOR competitive reactions). As a result, 86 % H₂O₂ production selectivity is achieved in SnO_{2-x}/BiVO₄ photoanode. Please refer to Ref. R4.

Fig. R1. (a) The surface thermodynamic processes of O_2 evolution on $BiVO_4$ (111), SnO_2 (110), and SnO_{2-x} (110). (b) Free energy diagram of water oxidation plotted at $U= 1.77$ V vs RHE on $BiVO_4$ (111), SnO_2 (110), and SnO_{2-x} . Reproduced from Ref. R4. (This is for reviewers only)

For the cathodic part, the selective $2e^-$ ORR mechanism for H_2O_2 evolution over Mo-SACs/mrG was also confirmed by both theoretical calculations with experimental measurements. The oxygen functional groups (OFGs) bridged single metal atoms with $M-O_x-C$ moieties show a universal overwhelming preference to form H_2O_2 as a result of the downhill reaction. As presented in Fig. R2a, different kinds of Mo-O bonds of $-O_3Mo$, $-O_2Mo$ and $-COOMo$ are investigated to explore the catalytic activity. Accordingly, the

O_2^* , OOH^* and O^* intermediates associated with the two-electron and four-electron pathways of the ORR over the $-O_3Mo$ single moiety are shown in Fig. R2b, and the corresponding free-energy diagrams are displayed in Fig. R2c, showing a common tendency in which OOH^* preferentially form H_2O_2 as a result of the downhill reaction, whereas the formation of $O-OH_2^*$ is an uphill reaction for all cases. Please refer to Ref. R5.

Fig. R2. (a) Models of different moieties: $-O_2M$, $-O_2M$ and $-COOM$. (b) Differential electron density of the intermediates of the two-electron and four-electron pathways of the ORR over the single $-O_3M$ moiety. (c) Free energy of the 2e⁻ and 4e⁻ ORRs plotted at $U= 0$ V and $pH=7$ for the active moiety of $-COOMo$, $-O_2Mo$ and $-O_3W$. Reproduced from Ref. R6. (This is for reviewers only)

These facts have been mentioned in our revised manuscript.

In this work, we presented a rationally and ingeniously designed artificial leaf by using the above two materials and further developed a self-cycled photo-Fenton-like system by

introducing $\text{Mn}^{2+}/\text{Mn}^{4+}$ redox, which achieved a world record of solar to H_2O_2 for pollution degradation. The improved strategy and novelty can be summarized as follows:

1. By using $\text{SnO}_{2-x}/\text{BiVO}_4/\text{WO}_3$ instead of $\text{SnO}_{2-x}/\text{BiVO}_4$, the onset potential was cathodically shifted to successfully couple with a cathodic reaction.
2. The $\text{PTFE}@Mo\text{-SACs}/\text{mrG-GDE}$ instead of $\text{Mo-SACs}/\text{mrG}$ remarkably improved the reaction kinetics
3. The $\text{Mn}^{2+}/\text{Mn}^{4+}$ redox couple was put forward to catalyze the on-site H_2O_2 product for ROSs generation, and further degrade various dyes with 30-day stability, and the designed device can be scaled to 70 cm^2
4. The indirectly generating ROSs by photochemical approach opened a new way for highly efficient and stable pollution degradation

1#5: It is important to include a video on the reactor setup for the solar-to- H_2O_2 conversion for both lab-scale and prototype devices.

Our Response: Thanks for your kind advice. In this work, we focused on in-situ utilization of on-site generated H_2O_2 for dye degradation, the reactor setups of both lab-scale and prototype devices are not considered yet. We hope the prototype device can be developed anywhere in the future. Alternatively, in our revised manuscript, the video of the $7\times 10\text{ cm}^2$ artificial leaf (using the rule as reference) for the effective degradation of a 200 mL solution containing 5 ppm rhodamine B (Rh. B), 5 ppm methylene blue (MB) and 5 ppm NP was provided. The dyes solution become colorless and transparent within 70 min (compressed into 33s in the video) with only light and oxygen supplementation, indicating the successful removal of the dyes.

This video is included as the supplementary movie.

1#6: The authors should provide a comparison table to compare the present results with the state-of-the-art catalytic performance of H_2O_2 production. This is extremely important to advance state-of-the-art research in the realm of renewable H_2O_2 production.

Our Response: Thanks for your kind advice. In order to evaluate the catalytic performance of our device, we have made a comparison table and updated the summary of PEC H_2O_2 production with a state-of-the-art PEC device.

Electrode Component	H ₂ O ₂ production rate (μmol*cm ⁻² *min ⁻¹) With bias (bias-free)	STH efficiency (%) With bias (bias-free)	With/without PV device	Two-electrode/three-electrode	Applied Bias (V vs. RHE)	Year	Reference
SnO _{2-x} /BiVO ₄ /WO ₃ Mo-SACs/mrG	2.63(1.76)	2.18(1.46)	no	both	1.76(Bias-free)	2022	This work
SnO _{2-x} /BiVO ₄	0.99	1.87	no	Three-electrode	1.23	2020	23
RuO ₂ /TNR AQ/Graphite	(0.35)	(0.66)	no	Two-electrode	Bias-free	2021	39
P-Mo-BiVO ₄ AQ-CNT/C	0.66(0.16)	1.25(0.30)	no	both	1 (Bias-free)	2020	40
AQ-DSPECs	0.5	0.95	no	Three-electrode	0.13	2020	41
NiFeO _x -BiVO ₄ PTTh	0.4	0.75	no	Three-electrode	0.65	2020	42
WO ₃ /BiVO ₄	0.084	0.16	no	Three-electrode	0.56	2016	43
m-WO ₃ carbon paper-Co ^{II/III} (Ch)	(0.2)	(0.38)	no	Two-electrode	Bias-free	2016	44
Al ₂ O ₃ /WO ₃ /BiVO ₄	1.0	1.89	no	Three-electrode	1.5	2017	45
BiVO ₄ carbon	(0.48)	(0.91)	no	Two-electrode	Bias-free	2018	46
WO ₃ /BiVO ₄ carbon	(0.13)	(0.25)	no	Two-electrode	Bias-free	2017	47
FeOOH-BiVO ₄ carbon paper-Co ^{II/III} (Ch)	(0.2)	(0.38)	no	Two-electrode	Bias-free	2017	48
PSK/O-BP α-NiFeO _x /CP	(1.74)	(1.41)	(MAPbI ₃) perovskite solar cell	Two-electrode	Bias-free	2021	49

This Table is included as supplementary Table 1.

1#7: The authors observed no deterioration after one month showing the robustness of the artificial leaf. It is important to further understand and comprehensively characterize the physicochemical properties of the spent catalysts.

Our Response: Thanks for your nice question. Our device shows good stability after stability measurement within one month as presented in Supplementary Fig. 31 in our original manuscript.

Indeed, the BiVO₄ composition is not completely stable during PEC measurements, as the BiVO₄ can be dissolved by the surface accumulated holes in an aqueous solution (Ref. R6). As a result, pure BiVO₄ presented poor stabilities, as demonstrated in Supplementary Fig. 13 in our original manuscript. Modified heterostructure of BiVO₄/WO₃ might facilitate the charge carrier transport and relieve the serious hole accumulation in the surface. More importantly, the surface coated SnO_{2-x} is believed to

act as a protection layer to suppress photo-corrosion of BiVO_4 in the bicarbonate solution, as demonstrated in Supplementary Fig. 13 in our original manuscript. To further understand the physicochemical properties of the photoanode and cathode, we investigated its morphology/composition/chemical state changes during the long-term testing.

The SEM image of photoanode after long-term stability testing shows the morphology of $\text{SnO}_{2-x}/\text{BiVO}_4/\text{WO}_3$ is well-maintained (Fig. R3a). Meantime, HR-TEM also demonstrates a conformally and stable SnO_{2-x} overlayer on BiVO_4 (Fig. R3b). Moreover, EPR measurement evidenced the maintenance of the oxygen vacancies of the SnO_{2-x} layer after stability tests (Fig. R3c), which indicated that the passivation layer of SnO_{2-x} seems to be responsible for the good stability of $\text{SnO}_{2-x}/\text{BiVO}_4/\text{WO}_3$.

Fig. R3: (a) SEM of $\text{SnO}_{2-x}/\text{BiVO}_4/\text{WO}_3$ photoanode after long-term stability tests. (b) HR-TEM of $\text{SnO}_{2-x}/\text{BiVO}_4/\text{WO}_3$ photoanode after long-term stability tests. (c) EPR spectra of $\text{SnO}_{2-x}/\text{BiVO}_4/\text{WO}_3$ photoanode before and after long-term stability tests. (This Figure is included as an inset of Supplementary Fig. 31 in the revised manuscript).

Furthermore, we performed the XPS investigations on the composition of the surface and interfaces of $\text{SnO}_{2-x}/\text{BiVO}_4/\text{WO}_3$ after the long-term testing. As shown in Fig. R4a, the O1s peak shows detectable changes in intensity and position, whereas a slight shift of the V 2p towards lower binding energy and reduced intensity can be observed after long-term testing. The phenomenon can be tracked to the photo-charging of BiVO_4 (Ref. R7), by which the V^{5+} state was somehow reduced to V^{4+} . Nevertheless, the Sn 3d peaks were determined to be stable during PEC testing (Fig. R4b).

Fig. R4: Comparisons of XPS results of $\text{SnO}_{2-x}/\text{BiVO}_4/\text{WO}_3$ before and after long-term testing. (a) V 2p and O1s and (b) Sn 3d (This Figure is included as Supplementary Fig. 32 in the revised manuscript).

On the other hand, the Mo-SACs-mrG-GDE also shows good stability with well-maintained singly dispersed atoms (Fig. R5a) and quasi-nanoarray aerophilicity areas (Fig. R5b). The modified oxygen-diffusion-benefit structure is supposed to contribute to good stability since the insufficient oxygen supplement during large current density would damage the catalysts.

Fig. R5: (a) SEM of Mo-SACs/mrG-GDE cathode after long-term stability tests. (b) HAADF-STEM of $\text{SnO}_{2-x}/\text{BiVO}_4/\text{WO}_3$ after long-term stability tests. (This Figure is included as Supplementary Fig. 33 in the revised manuscript).

1#8: Minor: Carefully check the entire manuscript for grammatical errors. For example, in line 63-65, “In recent years, bicarbonate (HCO_3^-) was found that not only can react with H_2O_2 to form HCO_4^- with high reactivity, but also can act as an auxiliary for water oxidative H_2O_2 evolution under photoelectrochemical (PEC) condition...”, and many more.

Our Response: Thanks for your kind advice. The manuscript has been carefully polished by Springer Nature Author Services.

Reviewer #2: This manuscript reported unassisted solar H₂O₂ production by combining SnO₂/BiVO₄/WO₃ || PTFE@Mo-SAC/mrG-GDE in a bicarbonate-containing electrolyte with a high production rate of 0.77 μmol/min cm⁻² and achieved efficiency of 1.46%. This combined system showed 150 % faradaic efficiency from both the photoanode and cathode. When demonstrated the large-scale artificial leaf, unassisted current of 10.7 mA with a production rate of 5.17 μmol min⁻¹ is achieved by this system. Furthermore, the successful application was found when such a system was applied for the degradation of 4-nitrophenol in one-pot self-cycled wastewater treatment. Overall, the manuscript introduces a novel concept for the dual application of artificial photosynthesis of H₂O₂ and greener wastewater treatment. The manuscript is a well-written interesting paper and is publishable in “nature communications” after the modifications considering comments below.

Our Response: We thank you very much for your positive evaluation of our work. As will be shown below, we have polished our manuscript towards higher quality through significant modifications based on your suggestions.

2#1: In the energy level diagram, in Fig. 2, the authors should mention the reference if the values of energy are not measured by themselves or mention the name of equipment used for measuring the values.

Our Response: Thanks for your kind advice. We have added the references for the band gap values in the revised manuscript.

2#2: In Fig. 5a, the intersection point of both the electrodes exhibited 2.07 mA cm⁻², whereas the current was reduced to 1.56 mA cm⁻² in Fig. 5b. The authors need to explain this difference and need to measure LSV at lower scan rates too to exactly determine the intersection current.

Our Response: Thanks for your wise advice. It is wise to slow the scan rates during the LSV measurements to minimize the impact of the non-Faradic current. Therefore, we have retested the LSV of both cathode and photoanode with various scanning rates of 2, 5 and 10 mV/s (original scan rate). As shown in Fig. R6, a slightly higher current density is

observed at a higher scanning rate. In our revised manuscript, we have unified the scanning rate to 2 mV/s, which achieves a theoretical intersection current of 1.64 mA/cm² (0.61 V vs. RHE) that is closed to the value measured in the two-electrode system. The reason for the slight difference between theoretical value and the actual bias-free current might be attributed to the polarization that occurs at the counter electrode (Ref. R8) in the two-electrode system compared to the three-electrode system.

Fig. R6: Coupled LSV scan of the SnO_{2-x}/BiVO₄/WO₃ photoanode and PTFE@Mo-SACs/mrG-GDE cathode with different scanning rates.

2#3: The authors need to add the error bars in Fig. 3g, Fig. 4b, Fig. 4e, Fig. 5d, and Fig. 6d to increase the reliability of data.

Our Response: Thanks for your kind advice. Error bars have been added.

2#4: Some minor errors. This includes:

- Fig. 5e cm² on the Y1 axis, Fig. 6 legends 7 cm² is indicated instead of 70 cm² and also, in Fig. 6b.
- Citation of Fig. in line 88.
- Line 140, the subscript of production rate.
- Numerical value of H₂O₂ selectivity in line 222.
- mv/dec to mV/dec.
- On page 8, line 157, there is no end bracket next to ‘Supplementary Fig. 6’
- From method section, line 399, authors synthesized WO₃ at 110°C for 40 min, but from Table S1, optimized WO₃ is synthesized at 110°C for 60 min. Make them clear.
- From Fig. 5c, 5d and 6c, change time unit from capital ‘S’ to small ‘s’

Our Response: Thanks for your kind advice. We are sorry for these mistakes caused by our carelessness. All the format and statements are carefully corrected and note that The geometric area of artificial leaf used in Fig. 6 is 7 cm²).

Reviewer #3: In this manuscript, the author designs a self-cycled photo-Fenton-like system for real-time wastewater purification. This is an interesting study. However, in terms of the mechanistic discussion, the authors do not consider the role of reactive species other than hydroxyl radicals at all. Therefore I don't think it can be published.

Our Response: We thank you very much for your evaluation of our work. The questions and suggestions raised by you are extremely important and helpful, which make us deep thought, thereby improving the quality of our work in the revision. We agree with you that there are other reactive oxygen species (ROSs) such as $^1\text{O}_2$ and $\text{O}_2^{\cdot-}$ based on our new investigations. The related statements have been carefully corrected.

3#1: 1. It is well known that the removal of pollutants may be the result of the combined action of multiple active species. However, only the role of hydroxyl radicals is explored in the manuscript, and the possible roles of other reactive species such as $^1\text{O}_2$ and $\text{O}_2^{\cdot-}$ are not considered.

Our Response: Thanks for your nice advice. We agree with you that there might be various ROSs such as $\cdot\text{OH}$, $^1\text{O}_2$ and $\text{O}_2^{\cdot-}$ et al. Therefore, we re-evaluated our self-cycled AOPs system with specific concerns over other ROSs included $^1\text{O}_2$ and $\text{O}_2^{\cdot-}$. As shown in Fig. R7, the signal of characteristic peaks of $\text{O}_2^{\cdot-}$ appears in the electron paramagnetic resonance (EPR) spectrum of the aqueous solution with co-existence of HCO_3^- , H_2O_2 , and $\text{Mn}(\text{II})$, which are obvious than that of the $\text{HCO}_3^-/\text{H}_2\text{O}_2$ or $\text{H}_2\text{O}_2/\text{Mn}(\text{II})$, indicating that the $\text{Mn}(\text{II})$ and HCO_3^- can also notably activate H_2O_2 into $\text{O}_2^{\cdot-}$. Besides, only a slight $^1\text{O}_2$ signal is observed in the EPR signal spectrum of the aqueous solution with the co-existence of HCO_3^- , H_2O_2 , and $\text{Mn}(\text{II})$ while barely appearing the $\text{HCO}_3^-/\text{H}_2\text{O}_2$ or $\text{H}_2\text{O}_2/\text{Mn}(\text{II})$. Based on the above evidence, it is believed that multi-species of ROSs including the $\cdot\text{OH}$, $\text{O}_2^{\cdot-}$ and $^1\text{O}_2$ can be activated during the self-cycled AOPs system, which is close to previous studies (Ref. R9). The related statements are corrected in our revised manuscript.

Fig. R7: EPR response of (a) $O_2^{\bullet-}$ and (b) 1O_2 in the AOP system with/without Mn^{2+} or HCO_3^- . The interfere peak in (a) except for the $O_2^{\bullet-}$ is caused by the H_2O molecule in the aqueous solution (the appropriate electrolyte should be the methanol). (This Figure is included as Supplementary Fig. 1 in the revised manuscript).

3#2: As mentioned before, the authors constructed a relatively complex system including catalysts, O_2 , HCO_3^- , $Mn(II)$ and $Mn(III)$, etc. For such a system, the contribution of multiple active species to pollutant degradation and the possible mutual transformation between them should be systematically considered. In fact, there may be 1O_2 and $O_2^{\bullet-}$ intermediates for the formation of H_2O_2 . Therefore, I do not think it is reliable to attribute removal of pollutants entirely to hydroxyl radical.

Our Response: Thanks for your nice advice. We agree with you that there are several ROSs involved in the degradation of the dye in our AOP system according to the new evidence presented in Q3#1. We have been carefully investigated related references about the metal ion-assisted bicarbonate activated hydrogen peroxide (BAP) system to reconsider the contribution of the ROSs to pollutant degradation. Actually, there is no generally accepted conclusion about the H_2O_2 activation pathway in the Mn-assisted BAP system because of the complicated mutual interaction between the different ROSs. The potential interaction between ROSs related to our AOP system are presented here (Ref. R10):

which is consistent with our new evidence presented in Q3#1. Above all, we believe that a variety of ROSs is responsible for the oxidation process. The related descriptions in the manuscript have been corrected.

3#3: The author proposes that the generation process of H_2O_2 is $\text{O}_2 + 2\text{H}_2\text{O} \rightarrow 2\text{H}_2\text{O}_2$. If the external energy output is sufficient to drive this reaction, holes may directly oxidize $\text{H}_2\text{O}/\text{OH}^-$ to $\cdot\text{OH}$. Did the author consider this process?

Our Response: Thanks for your nice advice. Photo-induced holes directly oxidize the H_2O to $\cdot\text{OH}$, which is known as $1e^-$ WOR in the following equation:

The fact was well-demonstrated in our previous work by using $\text{SnO}_{2-x}/\text{BiVO}_4$ photoanode, please refer to Ref.R4. However, the amount of directly formed $\cdot\text{OH}$ is much less than that of H_2O_2 . Therefore, the obtained high efficiency for pollution degradation can be mainly ascribed to the high efficiency of solar to H_2O_2 conversion and then ROSs formation.

Reference:

1. Hu S. Membrane-less photoelectrochemical devices for H_2O_2 production: efficiency limit and operational constraint. *Sustain. Energy Fuels* **3**, 101-114 (2019).
2. Xu J, *et al.* Organic wastewater treatment by a single-atom catalyst and electrolytically produced H_2O_2 . *Nat. Sustain* **4**, 233-241 (2020).
3. Yu J, Li BQ, Zhao CX, Liu JN, Zhang Q. Asymmetric Air Cathode Design for Enhanced Interfacial Electrocatalytic Reactions in High-Performance Zinc-Air Batteries. *Adv Mater* **32**, e1908488 (2020).
4. Zhang K, *et al.* Near-Complete Suppression of Oxygen Evolution for Photoelectrochemical H_2O Oxidative H_2O_2 Synthesis. *J. Am. Chem. Soc.* **142**, 8641-8648 (2020).
5. Dong C, *et al.* Precise Synthesis of Single atom (Mo, W, Nb) Coordinated with

- Oxygen Functional Groups of Graphene Oxide for Stable and Selective Two-electron Oxygen Reduction in Neutral Media. *J. Mater. Chem. A*, 10.1039/D1TA10529A (2022).
6. Lee DK, Choi K-S. Enhancing long-term photostability of BiVO₄ photoanodes for solar water splitting by tuning electrolyte composition. *Nat. Energy* **3**, 53-60 (2017).
 7. Trzeźniewski BJ, *et al.* Near-complete suppression of surface losses and total internal quantum efficiency in BiVO₄ photoanodes. *Energ Environ Sci* **10**, 1517-1529 (2017).
 8. Hodes G. Photoelectrochemical Cell Measurements: Getting the Basics Right. *J. Phys. Chem. Lett.* **3**, 1208-1213 (2012).
 9. Meng ZH, *et al.* Formation and Oxidation Reactivity of MnO₂(+)(HCO₃(-))_n in the Mn(II)(HCO₃(-))-H₂O₂ System. *Inorg. Chem* **59**, 3171-3180 (2020).
 10. Lin JM, Liu M. Chemiluminescence from the decomposition of peroxymonocarbonate catalyzed by gold nanoparticles. *J. Phys. Chem. B.* **112**, 7850-7855 (2008).

Reviewers' comments:

Reviewer #1 (Remarks to the Author):

The authors have thoroughly revised the manuscript based on the comments raised by all reviewers. The novelty of the work is high. The manuscript reaches the requirement for consideration in Nature Communications. I am happy with the present revision.

Reviewer #2 (Remarks to the Author):

Authors have addressed the comments, although authors forgot to update the error bars in the following comment.

Revised manuscript is now appropriate to be published in this journal.

1. The authors should add additional experiments data to add the error bar in Fig. 3g, Fig. 4b, Fig. 4e, Fig. 5d, and Fig. 6d.

Reviewer #3 (Remarks to the Author):

I again reviewed the revised manuscript with the author's response. I think their reply could not answer the problems noted.

1. The authors only investigated the signal intensities of $^{1}O_2$ and O_2^- in different systems through EPR, which is far from enough to explain the reaction mechanism. The authors did not even study the contribution of the generated $^{1}O_2$ and O_2^- converted from H_2O_2 or O_2 to pollutant removal, which is the most basic aspect.

2. The H_2O_2 generation pathway ($O_2 + 2H_2O \rightarrow 2H_2O_2$) proposed by the authors is unreliable because they did not verify other possible H_2O_2 generation pathway. In fact, both $^{1}O_2$ and O_2^- may be intermediates in the formation of H_2O_2 (Angew. Chem. Int. Ed. 2020, 59, 2–23; Applied Catalysis B: Environmental 250 (2019) 408–418). It's worth emphasizing that I already mentioned explicitly in

the first round of review (In fact, there may be 1O_2 and O_2^- intermediates for the formation of H_2O_2). However, the author does not answer any of it, but cites the literature on H_2O_2 decomposition, which is not relevant to my previous comment.

3. For 1#4 proposed by reviewer 1, the author also did not answer directly and made corresponding revisions. Reviewer 1 proposed that the content of DOS should be added. However, the author did not perform corresponding theoretical calculations, but cited "seemingly similar" literature results. The corresponding theoretical calculations should be carried out for different reaction systems, otherwise it is meaningless. Moreover, the results of the literature cited by the authors were simulations of reaction pathway, which were completely different from the DOS calculations proposed by reviewer 1.

3. Before the reaction, there are only catalysts, O_2 , HCO_3^- and $Mn(II)$ in the reaction system, and there is no H_2O_2 , because H_2O_2 is generated after the reaction starts. However, the authors did not consider the case where only HCO_3^- and $Mn(II)$ exist when using EPR to study the active species in the reaction, but the case where HCO_3^- , $Mn(II)$ and H_2O_2 exist. Therefore, I must say that the EPR done is unconvincing to study the mechanism of the current reaction system.

4. The author believes that ROS such as 1O_2 and O_2^- are converted from H_2O_2 only through EPR experiments, which is unconvincing. It is well known that O_2 can be converted to O_2^- or 1O_2 , and O_2^- can also be converted to 1O_2 . Moreover, ROS directly converted from O_2 may also directly participate in the degradation of pollutants before generating H_2O_2 , which is also a key point that the authors did not consider. Until this critical point is verified, the author's proposed "Self-cycled photo-Fenton-like system" is an unreliable conclusion.

5. The specific roles of $Mn(II)$ and HCO_3^- on the generation and interconversion of ROS are also very important. In some similar studies, the researchers only added one substance to the traditional reaction system, and used at least four different experimental methods to systematically explore the reaction mechanism (*Angewandte Chemie*, 133(6), 2939-2944; *Environ. Sci. Technol.* 2019, 53, 9725–9733). The reaction system proposed by the author is obviously more complex, and it is more necessary to systematically study the reaction mechanism, rather than simply obtaining an unreliable conclusion through EPR.

6. For 3#3, the reaction system of this work is quite different from that of ref 4, and the previous conclusions do not apply to this work.

In general, the author did not directly answer or directly ignored the questions raised by me and other reviewers, and this work has very serious problems in the reaction mechanism. Therefore, I do not think this work is suitable for publication.

1 **Response to the Comments on Nature Communications**

2 **Manuscript ID: NCOMMS-22-52128A**

3

4 **Dear Reviewers:**

5 We sincerely thank for your valuable comments. In our previous manuscript, we focused
6 on materials, device configuration and performances of our electrochemical devices. So,
7 Our description did not satisfy your question. In this revision, we have particularly focused
8 on the reaction mechanism issues during the ROS generation which is raised by reviewer
9 #3. Detailed and systematic researches have been performed and we believe the concerned
10 problems, which were the reason for rejection of this paper have been solved and the quality
11 of this manuscript is substantially improved. The added items are highlighted in red in the
12 main manuscript and the supplementary information. Following changes were made and
13 listed below:
14

15 **Reviewer(s)' Comments to Author:**

16 Reviewer #1: The authors have thoroughly revised the manuscript based on the
17 comments raised by all reviewers. The novelty of the work is high. The manuscript reaches
18 the requirement for consideration in Nature Communications. I am happy with the present
19 revision.

20 *Our Response: We thank you very much for your positive evaluation of our work.*

21

22 Reviewer #2: Authors have addressed the comments, although authors forgot to update the
23 error bars in the following comment. Revised manuscript is now appropriate to be
24 published in this journal.

25 *Our Response: We thank you very much for your positive evaluation of our work.*

26 2#1: The authors should add additional experiments data to add the error bar in Fig. 3g,
Fig. 4b, Fig. 4e, Fig. 5d, and Fig. 6d.

27 *Our Response: Thanks for your kind advice. We have added the references for the band*
28 *gap values in the re-submitted manuscript.*

29

30

31

32 Reviewer #3: I again reviewed the revised manuscript with the author's response. I think
33 their reply could not answer the problems noted.

34 **Our Response:** *Thanks for your valuable comments on our revised manuscript. Indeed, in*
35 *our manuscript, we focused on constructing artificial leaf to simultaneously meet anodic*
36 *water oxidative H₂O₂ and cathodic oxygen reductive H₂O₂ productions, and demonstrated*
37 *a cycled system of in-situ using those formed H₂O₂ by organic dye degradation experiment.*
38 *The questions and suggestions raised by you are central to the degradation mechanism,*
39 *including the ROS role in the degradation processes. We are sorry for our previous*
40 *responses that did not make you satisfy. In this revised manuscript, we have systematically*
41 *extended our research to solve those problems.*

42 3#1: The authors only investigated the signal intensities of ¹O₂ and O₂⁻ in different
systems through EPR, which is far from enough to explain the reaction mechanism. The
authors did not even study the contribution of the generated ¹O₂ and O₂⁻ converted from
H₂O₂ or O₂ to pollutant removal, which is the most basic aspect.

43 **Our Response:** *Thanks for your wise suggestion. In our re-submitted manuscript, the*
44 *sacrificial agent experiments were conducted to investigate the main ROSs in the system*
45 *with and without the Mn²⁺ species. Different sacrificial agents, including tert-butanol (TBA,*
46 *a sacrificial agent for ·OH), p-benzoquinone (BQ, a sacrificial agent for ·O₂⁻) and 2, 2, 6,*
47 *6-tetramethylpiperidine (TEMP, a sacrificial agent for ¹O₂) have been applied for the*
48 *investigation [Ref. R1]. In the system without Mn²⁺ species, a barely decrease in the*
49 *degradation rate (from 30.2 % to 29.5 % at 90 min) is observed when TEMP is added,*
50 *while a small inhibitory effect on the degradation (from 30.2 % to 27.6 % at 90 min) is*
51 *exhibited in the presence of BQ, suggesting the more positive role of ·O₂⁻ than ¹O₂ in the*
52 *degradation process. Specifically, the removal effect is almost inhibited after the addition*
53 *of TBA, suggesting two reasons: 1) the ·OH is the main ROS during the degradation. 2)*
54 *the ·O₂⁻ and ¹O₂ might be converted by the ·OH in this system (**please see the reaction***
55 ***mechanism in detail in Q3#6**). On the other hand, the addition of BQ, TEMP and TBA*
56 *leads to a certain decrease in the degradation rate in the system containing Mn²⁺ species*
57 *(decreased from 90.5 % to 80.7 %, 71.5 % and 28.1% at 40 min for the addition of TEMP,*
58 *BQ and TBA, respectively). Notably, an effective removal rate after the addition of TBA is*

59 still observed in the presence of Mn^{2+} species (28.1 % at 40 min) compared to that without
60 Mn^{2+} species (0.2 % at 40 min), indicating that the different interconversion mechanisms
61 of ROSs between the system with and without Mn^{2+} species, which will be further evaluated
62 in Q3#6.

63
64 Fig. R1. Degradation effect under different radical scavengers on 4-nitrophenol (4-NP) in the self-cycled
65 photo Fenton-like system of (a) without Mn^{2+} and (b) with Mn^{2+} . Reaction conditions: 0.4 M $NaHCO_3$
66 electrolyte (15 ml) with 10 ppm 4-NP, concentration of Mn^{2+} in (b) is 0.4 ppm, O_2 -saturated and bubbled
67 all the time, AM 1.5 illumination. (This Figure is included as Supplementary Fig. 35 in the revised
68 manuscript).

3#2: The H_2O_2 generation pathway ($O_2 + 2H_2O \rightarrow 2H_2O_2$) proposed by the authors is unreliable because they did not verify other possible H_2O_2 generation pathway. In fact, both 1O_2 and O_2^- may be intermediates in the formation of H_2O_2 (Angew. Chem. Int. Ed. 2020, 59, 2–23; Applied Catalysis B: Environmental 250 (2019) 408–418). It's worth emphasizing that I already mentioned explicitly in the first round of review (In fact, there may be 1O_2 and O_2^- intermediates for the formation of H_2O_2). However, the author does not answer any of it, but cites the literature on H_2O_2 decomposition, which is not relevant to my previous comment.

69
70 **Our Response:** Thanks for your kind advice. First, we are sorry for the negligence of the
71 previous comments and the absence of the investigation/discussion about the potential
72 intermediate roles of both 1O_2 and $\cdot O_2^-$ during the H_2O_2 production. Herein, the H-cell
73 configuration that separates cathode and anode with a Nafion membrane was taken to
74 specifically investigate the roles of 1O_2 and $\cdot O_2^-$ during the H_2O_2 production in both

75 cathode and anode. For the cathode part, the scavenger experiments using BQ and TEMP
 76 over PTFE@Mo-SACs/mrG-GDE were performed. As shown in Fig. R2a and b, the
 77 amount of the H_2O_2 is invariant with respect to the concentration of both BQ and TEMP,
 78 indicating that the quenching of either 1O_2 or $\cdot O_2^-$ has no effect on the H_2O_2 formation
 79 during the ORR in the cathode part [Ref. R2]. On the other hand, similar experiment results
 80 were also observed in the anode part (Fig. R2c and d) and neither the quenching of 1O_2
 81 nor $\cdot O_2^-$ affect the generation of H_2O_2 during the WOR in the anode part. Based on the
 82 above results, it is believed that neither 1O_2 nor $\cdot O_2^-$ is the intermediate during the H_2O_2
 83 production from either 2e-ORR or 2e-WOR [Ref. R3].

84

85 Fig. R2. The effect of BQ and TEMP on the H_2O_2 production over cathode (a,b) and anode (c,d).
 86 Reaction conditions: 0.4 M $NaHCO_3$ electrolyte (50 ml for each cathode and anode in H-cell), BQ (0.10,
 87 0.20 and 0.30 mM), TPA (0.10, 0.20 and 0.30 mM), O_2 -saturated and bubbled all the time, AM 1.5
 88 illumination. (This Figure is included as Supplementary Fig. 34 in the revised manuscript).

3#3: For 1#4 proposed by reviewer 1, the author also did not answer directly and made corresponding revisions. Reviewer 1 proposed that the content of DOS should be added. However, the author did not perform corresponding theoretical calculations, but cited “seemingly similar” literature results. The corresponding theoretical calculations should be carried out for different reaction systems, otherwise it is meaningless. Moreover, the results of the literature cited by the authors were simulations of reaction pathway, which were completely different from the DOS calculations proposed by reviewer 1.

89

90 ***Our Response:*** *Thanks for your kind advice. First, we believed that the DOS calculations*
91 *proposed by reviewer 1 are related to the catalytic mechanism for anodic and cathodic*
92 *H₂O₂ formation, rather than electronic structure, since both reactions compete with 4-*
93 *electron reactions. Secondly, as we all know, the catalytic reaction mainly occurs on the*
94 *surface of the catalyst. In this work, the catalytic surfaces of anodic and cathodic reactions*
95 *are SnO_{2-x} and Mo-SACs/mrG, respectively, both are the same as the cited literature (Note*
96 *that the cited references are also our works). In detail, in the cited literature for*
97 *photoanode, we constructed the SnO_{2-x} layer coated BiVO₄ photoanode for PEC water*
98 *oxidative H₂O₂ evolution, the calculated surface for the water oxidative H₂O₂ process is*
99 *SnO_{2-x}. In this work, we constructed the SnO_{2-x} layer coated BiVO₄/WO₃ photoanode, the*
100 *calculated surface for the water oxidative H₂O₂ process should be SnO_{2-x}. The difference*
101 *between this work and the cited literature is the introduction of WO₃. As confirmed by SEM*
102 *and TEM images, WO₃ components were conformally coated by BiVO₄ and SnO_{2-x} in*
103 *sequence, which means that the WO₃ would not participate in the surface catalytic reaction.*
104 *In the cited literature for the cathode, the introduced PTFE was not the active site for*
105 *oxygen reduction reaction but forms a hydrophobic surface for easier accessing O₂. Indeed,*
106 *the introduced WO₃ in photoanode and PTFE in the cathode is to facilitate reaction*
107 *kinetics, including charge separation and lowering the on-set potential, for meeting the*
108 *requirements of anodic water oxidative H₂O₂ and cathodic oxygen reductive H₂O₂*
109 *evolutions.*

110

3#4: Before the reaction, there are only catalysts, O_2 , HCO_3^- and $Mn(II)$ in the reaction system, and there is no H_2O_2 , because H_2O_2 is generated after the reaction starts. However, the authors did not consider the case where only HCO_3^- and $Mn(II)$ exist when using EPR to study the active species in the reaction, but the case where HCO_3^- , $Mn(II)$ and H_2O_2 exist. Therefore, I must say that the EPR done is unconvincing to study the mechanism of the current reaction system.

111

112 **Our Response:** Thanks for your kind advice. As we know, HCO_3^- and Mn^{2+} are stable
113 anion and cation, respectively, they would not generate any radicals in general. Therefore,
114 in our previous manuscript, we didn't provide the measurement in the presence of HCO_3^-
115 and Mn^{2+} only. In this revised manuscript, control experiments were performed to
116 investigate the degradation effect with only HCO_3^- , Mn^{2+} (or O_2) in the reaction system.
117 Degradation experiments were evaluated under different circumstances (Fig. R3). It shows
118 that only the sample with an effective photoelectrochemical process activated by the
119 artificial leaf is able to decompose the pollutant. Therefore, the above results also suggest
120 that the ROSs are converted from the H_2O_2 provided in the self-cycled system rather than
121 directly from HCO_3^- and Mn^{2+} .

122

123 Fig. R3. The degradation effect of different compositions in the reaction system. Reaction conditions:
124 0.4 M $NaHCO_3$ electrolyte (15 ml) with 10 ppm 4-NP, concentration of Mn^{2+} is 0.4 ppm, AM 1.5
125 illumination. (This Figure is only for reviewer 3).

3#5: The author believes that ROS such as $^1\text{O}_2$ and O_2^- are converted from H_2O_2 only through EPR experiments, which is unconvincing. It is well known that O_2 can be converted to O_2^- or $^1\text{O}_2$, and O_2^- can also be converted to $^1\text{O}_2$. Moreover, ROS directly converted from O_2 may also directly participate in the degradation of pollutants before generating H_2O_2 , which is also a key point that the authors did not consider. Until this critical point is verified, the author's proposed "Self-cycled photo-Fenton-like system" is an unreliable conclusion.

126

127 ***Our Response:*** Thanks for your kind advice. Experiment results presented in Q3#2
128 suggested that $^1\text{O}_2$ and $\cdot\text{O}_2^-$ are not supposed to be the intermediates for H_2O_2 production
129 during the O_2 reduction in the cathode. Moreover, RRDE tests have shown that the main
130 product of O_2 reduction in the cathode is H_2O_2 (Fig. 4a). Above all, we believed that the
131 $^1\text{O}_2$ and $\cdot\text{O}_2^-$ are converted from the in-situ generated H_2O_2 in the self-cycled photo-like
132 system. On the other hand, the interaction of ROSs is important for understanding the
133 mechanism of the degradation reaction, and a more detailed discussion would be evaluated
134 in Q3#6.

3#6: The specific roles of Mn(II) and HCO_3^- on the generation and interconversion of ROS are also very important. In some similar studies, the researchers only added one substance to the traditional reaction system, and used at least four different experimental methods to systematically explore the reaction mechanism (Angewandte Chemie, 133(6), 2939-2944; Environ. Sci. Technol. 2019, 53, 9725–9733). The reaction system proposed by the author is obviously more complex, and it is more necessary to systematically study the reaction mechanism, rather than simply obtaining an unreliable conclusion through EPR.

135

136 ***Our Response:*** Thanks for your kind advice. First, KPi electrolyte with the same pH value
137 (adjusted by KOH) is taken to replace NaHCO_3 electrolyte to evaluate the role of HCO_3^-
138 in the self-cycled photo-like system. As shown in Fig. R4a, barely degradation effect is
139 observed in the absence of Mn^{2+} , even if H_2O_2 is effectively generated in the reaction
140 system (via 2e-ORR in cathode) (Fig. R4b), which indicates KPi cannot activate H_2O_2 into
141 ROS as the HCO_3^- does (Fig. 6d and Fig. S1). Furthermore, the degradation rate in KPi
142 electrolyte slightly raised to 5.4% at 40 min when Mn^{2+} is added, which is still much lower

143 *than that in HCO₃⁻ electrolyte with Mn²⁺ (89.5% at 40 min). That means the effective*
 144 *activation of H₂O₂ is largely suppressed in the absence of HCO₃⁻.*

145

146 Fig. R4. (a) The degradation effect in KPi electrolyte in the reaction system. (b) The H₂O₂ concentration
 147 in the self-cycled Fenton-like reaction system with different electrolytes. Reaction conditions: (a) 0.4
 148 M NaHCO₃ or 0.4 M KPi electrolyte with the same pH value (15 ml) with 10 ppm 4-NP, the
 149 concentration of Mn²⁺ is 0.4 ppm, O₂-saturated and bubbled all the time, AM 1.5 illumination in the
 150 degradation tests. (b) 0.4 M NaHCO₃ or 0.4 M KPi electrolyte with the same pH value (15 ml), O₂-
 151 saturated and bubbled all the time, AM 1.5 illumination in the degradation tests in the H₂O₂
 152 concentration measurement. (This Figure is included as Supplementary Fig. 36 in the revised
 153 manuscript).

154 *Thus, EPR technology was employed to further investigate the mechanism.*
 155 *Methanol and DMPO were added as capture agents to exclude the interface of ·OH for*
 156 *detecting ·O₂H [Ref. R4]. As shown in Fig. R5a, no signal of ·O₂H was recognized in the*
 157 *system, indicating ·O₂H is not generated as an intermediate in the system. The signal*
 158 *of ·O₂⁻ is unchanged after the addition of TEMP while disappeared after the addition of*
 159 *TBA, indicating that the ·OH is essential for the generation of ·O₂⁻ while ¹O₂ is not [Ref.*
 160 *R5]. (Fig. R5b). Besides, no obvious signal is observed in the detection of ¹O₂ in the*
 161 *absence of Mn²⁺ (Fig. R5c). Moreover, the addition of TEMP and BQ has no impact on ·OH*
 162 *(Fig. R5d). Therefore, the mechanism for the ROS generation could be given:*

167

168 Fig. R5. EPR response of (a) $\cdot\text{O}_2\text{H}$, (b) $\cdot\text{O}_2^-$, (c) ${}^1\text{O}_2$ and (d) $\cdot\text{OH}$ in the self-cycled photo-Fenton-like
 169 system (without Mn^{2+}) accompanied with the different radical scavenger. Reaction conditions: 0.4 M
 170 NaHCO_3 electrolyte (15 ml), O_2 -saturated and bubbled all the time, AM 1.5 illumination for 15 min.
 171 (This Figure is included as Supplementary Fig. 37 in the revised manuscript).

172 *On the other hand, the addition of Mn^{2+} in HCO_3^- electrolyte could largely*
 173 *increase the degradation rate by improving the generation of both ${}^1\text{O}_2$, $\cdot\text{O}_2^-$ and $\cdot\text{OH}$ (Fig.*
 174 *6d and Fig. S1). While the Mn^{2+} in KPi electrolyte shows a little effect of degradation (Fig.*
 175 *R4a), indicating that the complex between Mn^{2+} and HCO_3^- plays the main role in*
 176 *generating the ROS, which is also proved by the Fig. S1. EPR measurements were*
 177 *performed to further investigate the mechanism. Unlike in the HCO_3^- system, the addition*
 178 *of TBA has little impact on the generation of $\cdot\text{O}_2^-$ and ${}^1\text{O}_2$ during the EPR measurements,*
 179 *indicating that $\cdot\text{OH}$ is not essential for generating them in the $\text{Mn}^{2+}(\text{HCO}_3^-)_n$ system (Fig.*
 180 *R6a and 6b). However, the signal for ${}^1\text{O}_2$ disappeared after the addition of BQ, indicating*
 181 *${}^1\text{O}_2$ come from $\cdot\text{O}_2^-$ in the $\text{Mn}^{2+}-\text{HCO}_3^-$ system (Fig. R6a) [Ref. R6]. Moreover, the*

182 addition of TEMP and BQ has no impact on $\cdot\text{OH}$ (Fig. R6c). Above all, the ROS generation
 183 in the $\text{Mn}^{2+}-\text{HCO}_3^-$ system is given:

187 In the self-cycled system, the transformation of Mn^{IV} species into Mn^{II} species would be
 188 largely enhanced through eq(7) by the reduction of a cathode according to Fig. 1d and Fig.
 189 S4.

191

192 Fig. R6. EPR response of (a) ${}^1\text{O}_2$, (b) $\cdot\text{O}_2^-$, (c) $\cdot\text{OH}$ in the self-cycled photo-Fenton-like system (with
 193 Mn^{2+}) accompanied with the different radical scavengers. Reaction conditions: 0.4 M NaHCO_3 electrolyte
 194 (15 ml), the concentration of Mn^{2+} is 0.4 ppm, O_2 -saturated and bubbled all the time, AM 1.5
 195 illumination for 15 min. (This Figure is included as Supplementary Fig. 38 in the revised
 196 manuscript).

3#7: For 3#3, the reaction system of this work is quite different from that of ref 4, and the previous conclusions do not apply to this work.

(3#3: The author proposes that the generation process of H₂O₂ is O₂ + 2H₂O → 2H₂O₂. If the external energy output is sufficient to drive this reaction, holes may directly oxidize H₂O/OH⁻ to •OH. Did the author consider this process?)

197

198 **Our Response:** *Thanks for your nice question. According to the experiment results in the*
199 *Q3#2 and Q3#6, neither ¹O₂, •O₂⁻ nor •O₂H is the intermediate for the H₂O₂ production*
200 *during the WOR, indicating that the generation selectivity for •OH can be approximately*
201 *calculated by eq (9):*

202
$$FE\%(\cdot OH) = 1 - FE\%(O_2) - FE\%(H_2O_2) \quad (9)$$

203 *Therefore, the selectivity of •OH production in the photoanode will be evaluated via the H-*
204 *cell configuration companies with the O₂ quantity as a function of reaction time under*
205 *different applied bias. As shown in Fig. R7a, the H₂O₂ selectivity is almost invariant during*
206 *the photoelectrochemical process when the applied voltage is altered. Besides, Fig. R7b*
207 *suggests that there is barely O₂ evolution during the WOR in the photoanode, which is*
208 *consistent with our previous work. Above all, •OH selectivity with a similar value under*
209 *different external bias is presented in Fig. R7c, which is explained the reason for the quasi-*
210 *Femi level tuning in our previous work [Ref. R8].*

211 *As expected, the •OH can be generated by the direct hole oxidation process, the result*
212 *is consistent with Fig. R1a. We have added the discussion in our revised manuscript.*

213

214 Fig. R7. (a) H₂O₂ FE values, (b) O₂ quantity number and (c) ·OH FE values of SnO_{2-x}/BiVO₄/WO₃
 215 photoanode while taking PTFE@Mo-SACs/mrG-GDE as cathode under different applied potential
 216 under AM 1.5 illumination in 50 mL 0.4 M NaHCO₃ electrolyte.

217 Reference:

- 218 1. Ji J, Yan Q, Yin P, Mine S, Matsuoka M, Xing M. Defects on CoS_{2-x}: Tuning Redox Reactions
 219 for Sustainable Degradation of Organic Pollutants. *Angew. Chem. Int. Ed.* **60**, 2903-2908 (2021).
- 220 2. Zhao S, Zhao X. Insights into the role of singlet oxygen in the photocatalytic hydrogen peroxide
 221 production over polyoxometalates-derived metal oxides incorporated into graphitic carbon
 222 nitride framework. *Appl. Catal. B* **250**, 408-418 (2019).
- 223 3. Hou H, Zeng X, Zhang X. Production of Hydrogen Peroxide by Photocatalytic Processes.
 224 *Angew. Chem. Int. Ed.* **59**, 17356-17376 (2020).
- 225 4. Guo Z, *et al.* Single-Atom Mn-N₄ Site-Catalyzed Peroxone Reaction for the Efficient
 226 Production of Hydroxyl Radicals in an Acidic Solution. *J. Am. Chem. Soc.* **141**, 12005-12010
 227 (2019).
- 228 5. Yi Q, *et al.* Singlet Oxygen Triggered by Superoxide Radicals in a Molybdenum Cocatalytic
 229 Fenton Reaction with Enhanced REDOX Activity in the Environment. *Environ. Sci Technol.*
 230 **53**, 9725-9733 (2019).

- 231 6. Zhu L, *et al.* Designing 3D-MoS₂ Sponge as Excellent Cocatalysts in Advanced Oxidation
232 Processes for Pollutant Control. *Angew. Chem. Int. Ed.* **59**, 13968-13976 (2020).
- 233 7. Corby S, Rao RR, Steier L, Durrant JR. The kinetics of metal oxide photoanodes from charge
234 generation to catalysis. *Nat. Rev. Mater.* **6**, 1136-1155 (2021).
- 235 8. Zhang K, *et al.* Near-complete suppression of oxygen evolution for photoelectrochemical
236 H₂O oxidative H₂O₂ synthesis. *J. Am. Chem. Soc.* **142**, 8641-8648 (2020).

REVIEWERS' COMMENTS

Reviewer #4 (Remarks to the Author):

In this research article entitled “Self-cycled photo-Fenton-like system based on an artificial leaf with a solar-to-H₂O₂ conversion efficiency of 1.46%”, an effective solar-driven photo-Fenton-like system is successfully designed based on the elaborately improved PEC tandem cell/artificial leaf with a very high H₂O₂ production rate and solar-to-H₂O₂ conversion efficiency. Moreover, detailed experiments have been carefully performed to investigate the mechanism of the self-cycled system in sustainable environmental treatment.

We carefully considered the comments raised by Reviewer 3 and the responses from the authors. (1) We agree with the perspective of Reviewer 3 that the reaction involving multiple ROS is a complex process. In the revised manuscript, the authors responded professionally to the comments raised by Reviewer 3 through a series of controlled experiments. (2) We believe that the most important contribution of this manuscript is employing “artificial leaves” to sustainably and efficiently produce H₂O₂ in both photoanode and cathode without bias, which is very important for environmental purification. (3) According to the experimental results (Supplementary Fig. 35), •OH play a major role in this solar-driven photo-Fenton-like system, and other ROS (eg. O₂•⁻ and 1O₂) has little contribution to the pollutants degradation. Therefore, we believe that paying too much attention to the generation of other ROS (such as O₂•⁻ and 1O₂) is unnecessary and weakens the core innovation of this manuscript.

In conclusion, this is a very interesting article and I think that the authors satisfactorily reply to the concerns raised by previous Referees. Before publication in Nat. Commun., there are still two questions need to be further addressed.

1. In Fig. 5d, the H₂O₂ production rate shows a decrease along with the reaction time. However, the cathode and photoanode are rather stable according to the durability tests in Fig. S12 and Fig. S21, respectively. So what leads to unremarkable decay?

2. Compared to the largely different electrochemical performance of samples with various PTFE loading amounts, why the difference in the contacting angle presented in Fig. S18 seems not that distinct obviously.

Response to the Comments on Nature Communications

Manuscript ID: NCOMMS-22-52128B-Z

Dear Reviewers:

We would like to thank the reviewer for their careful reading and helpful comments. We revised the manuscript thoroughly according to the comments and we believe the concerned problems have been solved. The added items are highlighted in red in the main manuscript and the supplementary information. Following changes were made and listed below:

Reviewer(s)' Comments to Author:

Reviewer #4: In this research article entitled “Self-cycled photo-Fenton-like system based on an artificial leaf with a solar-to-H₂O₂ conversion efficiency of 1.46%”, an effective solar-driven photo-Fenton-like system is successfully designed based on the elaborately improved PEC tandem cell/artificial leaf with a very high H₂O₂ production rate and solar-to-H₂O₂ conversion efficiency. Moreover, detailed experiments have been carefully performed to investigate the mechanism of the self-cycled system in sustainable environmental treatment. We carefully considered the comments raised by Reviewer 3 and the responses from the authors. (1) We agree with the perspective of Reviewer 3 that the reaction involving multiple ROS is a complex process. In the revised manuscript, the authors responded professionally to the comments raised by Reviewer 3 through a series of controlled experiments. (2) We believe that the most important contribution of this manuscript is employing “artificial leaves” to sustainably and efficiently produce H₂O₂ in both photoanode and cathode without bias, which is very important for environmental purification. (3) According to the experimental results (Supplementary Fig. 35), •OH play a major role in this solar-driven photo-Fenton-like system, and other ROS (eg. O₂•- and ¹O₂) has little contribution to the pollutants degradation. Therefore, we believe that paying too much attention to the generation of other ROS (such as O₂•- and ¹O₂) is unnecessary and weakens the core innovation of this manuscript.

In conclusion, this is a very interesting article and I think that the authors satisfactorily reply to the concerns raised by previous Referees. Before publication in Nat. Commun., there are still two questions need to be further addressed.

Our Response: We thank you very much for your positive evaluation of our work.

4#1: In Fig. 5d, the H₂O₂ production rate shows a decrease along with the reaction time. However, the cathode and photoanode are rather stable according to the durability tests in Fig. S12 and Fig. S21, respectively. So what leads to unremarkable decay?

Our Response: Thanks for your nice question. The H₂O₂ production rate tested in Fig. 5d is carried out in the one-cell configuration composed of SnO_{2-x}/BiVO₄/WO₃ photoanode and PTFE@Mo-SACs/mrG-GDE cathode under bicarbonate electrolyte, so there are two possible reasons for a slight decrease of the measured H₂O₂ concentration compared to the theoretical values. The first one is that the generated H₂O₂ could be partly consumed by HCO₃⁻ via equation 1, and the consuming rate could be accelerated under higher H₂O₂ concentration based on Le Chatelier's principle. The other one is that the generated H₂O₂ in the cathode might be trapped/consumed in the porous carbonaceous substrate, which also leads to the slight deviation of values between the experiment and theoretical prediction.

4#2: Compared to the largely different electrochemical performance of samples with various PTFE loading amounts, why the difference in the contacting angle presented in Fig. S18 seems not that distinct obviously.

Our Response: Thanks for your nice question. It should be noticed that the substrate used in the PTFE@Mo-SACs/mrG-GDE cathode is hydrophobic carbon paper without a gas-diffusion layer (GDL), which suggests the substrate is hydrophobic at the macroscopic level (as revealed in the contacting angle tests in the Fig. S18). In our work, deliberate decoration at the substrate is carried out that the well-dispersed PTFE nanoparticles set up a confined hydrophobic area on the Mo-SACs/mrG nanosheets, which might not show an obvious impact on hydrophobicity in the macroscopic level due to the small size scale of the PTFE nanoparticles. However, more precise tuning of the hydrophobicity on the catalysts is supposed to enhance the mass-transfer ability as well as the conductivity compared to the normal GDL structure, which endows better performance during the oxygen reduction reaction for H₂O₂ production.